# Evolution of Alzheimer’s Disease Therapeutics: From Conventional Drugs to Medicinal Plants, Immunotherapy, Microbiotherapy and Nanotherapy

**DOI:** 10.3390/pharmaceutics17010128

**Published:** 2025-01-17

**Authors:** Emma Ortiz-Islas, Pedro Montes, Citlali Ekaterina Rodríguez-Pérez, Elizabeth Ruiz-Sánchez, Talía Sánchez-Barbosa, Diego Pichardo-Rojas, Cecilia Zavala-Tecuapetla, Karla Carvajal-Aguilera, Victoria Campos-Peña

**Affiliations:** 1Laboratorio de Neurofarmacologia Molecular y Nanotecnologia, Instituto Nacional de Neurología y Neurocirugía, Manuel Velasco Suárez, Mexico City 14269, Mexico; emma.ortiz@innn.edu.mx (E.O.-I.); crodriguez@innn.edu.mx (C.E.R.-P.); 2Laboratorio de Neuroinmunoendocrinología, Instituto Nacional de Neurología y Neurocirugía, Manuel Velasco Suárez, Mexico City 14269, Mexico; pedrovolvox@gmail.com; 3Laboratorio de Neuroquímica, Instituto Nacional de Neurología y Neurocirugía, Manuel Velasco Suárez, Mexico City 14269, Mexico; ruizruse@yahoo.com.mx; 4Laboratorio Experimental de Enfermedades Neurodegenerativas, Instituto Nacional de Neurología y Neurocirugía, Manuel Velasco Suárez, Mexico City 14269, Mexico; talisabar@hotmail.com (T.S.-B.); cztecua@yahoo.com.mx (C.Z.-T.); 5Departamento de Biomedicina Molecular, Centro de Investigación y de Estudios Avanzados del Instituto Politécnico Nacional, Mexico City 07360, Mexico; 6Programa Prioritario de Epilepsia, Instituto Nacional de Neurología y Neurocirugía, Manuel Velasco Suárez, Mexico City 14269, Mexico; diego.pichardo@uabc.edu.mx; 7Laboratorio de Nutrición Experimental, Instituto Nacional de Pediatría, Mexico City 04530, Mexico; karla_ca@yahoo.com

**Keywords:** Alzheimer’s disease, medicinal plants, nanotechnology, probiotics, fecal microbiota transplantation, miRNAs, immunotherapy

## Abstract

Alzheimer’s disease (AD) represents an escalating global health crisis, constituting the leading cause of dementia among the elderly and profoundly impairing their quality of life. Current FDA-approved drugs, such as rivastigmine, donepezil, galantamine, and memantine, offer only modest symptomatic relief and are frequently associated with significant adverse effects. Faced with this challenge and in line with advances in the understanding of the pathophysiology of this neurodegenerative condition, various innovative therapeutic strategies have been explored. Here, we review novel approaches inspired by advanced knowledge of the underlying pathophysiological mechanisms of the disease. Among the therapeutic alternatives, immunotherapy stands out, employing monoclonal antibodies to specifically target and eliminate toxic proteins implicated in AD. Additionally, the use of medicinal plants is examined, as their synergistic effects among components may confer neuroprotective properties. The modulation of the gut microbiota is also addressed as a peripheral strategy that could influence neuroinflammatory and degenerative processes in the brain. Furthermore, the therapeutic potential of emerging approaches, such as the use of microRNAs to regulate key cellular processes and nanotherapy, which enables precise drug delivery to the central nervous system, is analyzed. Despite promising advances in these strategies, the incidence of Alzheimer’s disease continues to rise. Therefore, it is proposed that achieving effective treatment in the future may require the integration of combined approaches, maximizing the synergistic effects of different therapeutic interventions.

## 1. Introduction

Alzheimer’s disease (AD) is an age-related neurodegenerative disorder occurring mainly in older adults, it is the most common form of dementia, and it is an incurable disease. It is estimated that over 55 million people worldwide suffer from AD and dementia, with the number of cases expected to reach 131 million by 2050 [1,2,3,4].

Clinical symptoms of AD include progressive cognitive impairment, short-term memory impairment, emotional disturbances, and aphasia, among others [5,6]. As the disease progresses, neuropsychiatric symptoms appear; there is an evident loss of language, loss of the ability to solve problems and abstract thinking, impaired orientation, and physical deterioration [5,7]. As a result, patients are unable to perform their normal tasks of daily living, gradually lose their mobility, become bedridden, and die. As the disease progresses, patients require full-time care, adding a heavy emotional and financial burden on families and society [8].

AD is considered a multifactorial disease, and its pathophysiology is complex. However, it has specific histopathological features in the brain: neuritic plaques (NP) composed of amyloid β (Aβ) and intracellular neurofibrillary tangles (NFTs) composed of hyperphosphorylated tau protein [9,10]. Under normal conditions, both proteins (Aβ and tau) are soluble, but under pathological conditions, they assemble into beta-folded structures, leading to neuronal death. Dysfunction of Aβ and tau alone can trigger the development of AD [11].

This work aims to provide an overview of the versatility and evolution of various therapeutic strategies, based on major pathophysiological theories, that have been investigated for the potential treatment of Alzheimer’s disease. This highly complex and multifactorial pathology demands diverse approaches. In this context, heterogeneous fields of knowledge such as medicinal plant research and nanotechnology are examined.

## 2. Pathophysiology of Alzheimer’s Disease

Given the complexity of AD’s etiology, several mechanistic theories have been proposed to explain the underlying causes [12,13,14]. The cholinergic theory states that loss of cholinergic and neurotransmitter neurons causes the development of AD [15,16,17,18,19]. The Aβ cascade theory points out that NP containing misfolded Aβ plays a decisive role in the pathogenesis of AD [20,21,22]. In this way, Aβ also drives tau-mediated neurodegeneration [23]. Tau theory proposes that abnormal phosphorylation and truncation of tau protein underlie the progression of AD [24,25,26]. Vascular dysfunction theory emphasizes the role of cerebral circulation and endothelium-mediated processes in the pathogenesis of AD [27,28,29]. The neuroinflammatory theory has pointed out that inflammation, far from being the result of activation by the presence of NP and NFT, contributes as much or more to the pathogenesis than the aggregates themselves [30,31]. The presence of NPs and NFTs contributes significantly to neuronal death and, consequently, to synaptic damage, resulting in loss of memory and cognition (Figure 1).

### 2.1. Amyloid Hypothesis

The amyloid cascade hypothesis, arguably the most accepted and studied of the theories proposed for the genesis of AD, was proposed by Hardy and Higgins in 1992. It postulates that deposition of Aβ, the major component of the plaques, is the causative agent of Alzheimer’s pathology and that the neurofibrillary tangles, cell loss, vascular damage, and dementia follow as a direct consequence of this deposition [32]. The extracellular accumulation of NP would cause pathological changes such as neurotoxicity and neuronal inflammation, leading to neuronal death.

The presence of amyloid precursor protein (APP) and presenilins (PSEN1/2) mutations, responsible for familial forms of AD, is the basis for most of the transgenic animal models developed to date (Table 1). The aim of developing these models is to elucidate the mechanisms of pathology and to test new therapeutic strategies. A comprehensive review of the main phenotypes developed has been recently described by Zhong et al. [33]. These mutations significantly alter the proteolytic processing of APP and increase the formation of longer peptides (42 amino acids), which have a greater capacity for aggregation.

In recent decades, it has been demonstrated that different forms of Aβ such as soluble oligomers, dimers, and insoluble aggregates, are capable of causing toxic effects and neuronal death in both in vitro and in vivo models [91,92,93]. The oligomeric forms can alter neurotransmitter uptake/release, synaptic plasticity, and cellular receptor localization, as well as cause memory deficits in animal models [94,95,96]. While the neurotoxic properties of Aβ are closely related to its degree of assembly [91], it is also important to note the nature of the targets with which it interacts. Its interaction with α7 nicotinic receptors, N-methyl-d-aspartate receptor (NMDAR), as well as a cellular prion protein, triggers glial activation, mitochondrial damage, oxidative stress, dysregulation of Ca^2+^ homeostasis, and altered axonal transport [96].

Although this hypothesis has been the basis for a great deal of progress in the study of AD, some authors argue that it should be reconsidered in light of currently available data. It is now known that both accumulation and deposition of Aβ, do not directly correlate with cognitive impairment [39,97,98]. In recent years, this theory has been widely challenged by the negative results of most clinical trials designed to evaluate the efficacy of anti-Aβ antibodies [97,98,99,100].

Although APP and Aβ play an important and fundamental role in the development of AD, it is important to remember that it is a complex, multifactorial, and multigenic disease.

### 2.2. Tau Pathology of Alzheimer’s Disease

Normally, tau proteins play a fundamental role in maintaining the cytoskeleton of neuronal cells. This process involves not only the maintenance of neuronal structure but is also closely linked to the transport of vesicles and organelles in and out of the cell. Correct neuronal function depends on the stability of microtubules, which must be rigid enough to maintain neuronal shape and flexible enough not to interfere with brain function [101,102,103,104]. Recently, several studies have suggested that Tau may be involved in embryonic development, long-term potentiation [105], long-term depression [106,107], and metabolic rate depression in hibernating animals [108]. Tau also has a role in the protection of genomic DNA and RNA. Under oxidative or hyper-thermic stress conditions, the accumulation of dephosphorylated Tau in the nucleus of neurons is induced and necessary to protect DNA [109].

The role of Tau in the stability of the cytoskeleton is achieved through its interaction with tubulin via the microtubule-binding domain, which is located at the amino terminus of the protein. There are six isoforms in the CNS, each containing three or four repeating domains (3R/4R) and with or without 1 or 2 N-terminal inserts [110,111,112].

Under pathological conditions, hyperphosphorylation and truncation processes disrupt the normal function of Tau. These modifications prevent the binding of Tau protein to microtubules and destabilize the microtubule network [103,113,114]. This leads to axonal disarray, loss of axonal transport, and thus, loss of communication between neurons and synaptic loss, resulting in a neurodegenerative process [111,112,115,116].

Tau is a highly soluble protein in normal conditions. However, in pathological conditions, tau self-assembles into insoluble structures known as paired helical filaments (PHFs). It has been postulated that altered hyperphosphorylation of tau leads to changes in its function that favor its self-assembly into PHFs. Thus, Tau hyperphosphorylation theory suggests that the presence of hyperphosphorylated forms of tau is the primary event triggering its auto-aggregation in PHF and subsequent NFT formation [117,118].

This neurofibrillary pathology, which causes neuronal death and synaptic loss, is closely related to the cognitive decline observed in patients, suggesting their role in AD development.

### 2.3. Lipid Dysregulation in Alzheimer’s Disease

In recent years, abnormalities in the metabolism of cholesterol, sphingolipids, glycerophospholipids, and phosphatidylinositol 4,5-bisphosphate (PIP2) have been shown to play an important role in AD development by promoting Aβ (NP) and tau (NFT) deposition [15,19]. Similarly, it has been suggested that altered lipid metabolism may play an indirect role in causing Alzheimer’s disease [119,120]. The processing that leads to the formation of the Aβ peptide occurs at the membrane level. Thus, over-regulation of lipids, particularly cholesterol, significantly increases peptide formation by facilitating the binding of APP to β- and γ-secretase [121,122,123]. High cholesterol levels have also been shown to favor an oxidative environment induced by Aβ peptides, leading to mitochondrial and lysosomal dysfunction, changes that culminate in neuronal death [123,124,125]. It has been proposed that the accumulation of free fatty acids (FFAs) observed in the brains of Alzheimer’s disease patients leads to the accumulation of toxic lipid intermediates. These intermediates induce alterations in mitochondrial function and increase oxidative stress and lipid peroxidation [126,127,128].

Hashemi et al. demonstrated that free cholesterol, as well as that present in the membrane, not only accelerates the process of Aβ aggregation but is also capable of producing aggregates more rapidly and of significantly larger sizes. The aggregates formed in the bilayer can in turn separate and accumulate. The dissociation of the membrane is accelerated by the presence of cholesterol, demonstrating that the Aβ-lipid interaction is an important factor in the process of assembly as well as NP formation [15]. Similarly, the accumulation of cholesterol in the membrane can alter cell signaling pathways that lead to tau phosphorylation. Increased cholesterol levels have the ability to increase the activity of GSK3β as well as other types of kinases, favoring tau phosphorylation [120,129,130].

### 2.4. Inflammation in Alzheimer’s Disease

The role of chronic inflammation in the development of several neurodegenerative diseases has been observed in the last decade. In AD, glial cells are known to be responsible for regulating the inflammatory process within the CNS. During the progression of the disease, the activation of astrocytes and microglia is markedly increased in response to the accumulation of Aβ peptides and tau. These cells release pro-inflammatory cytokines into the brain, increasing BBB permeability and thereby recruiting immune cells from the periphery [131,132,133].

Microglia cells are innate immune response components that develop from yolk sac progenitors and migrate to the brain [134,135]. After postnatal development, brain resident microglia have the ability to regulate their population size through their self-renewal capacity [135,136,137]. Their function is related to homeostasis, development, differentiation, metabolism, synaptic plasticity, phagocytosis, immune regulation, and neuronal survival [137,138,139,140]. Modulation of their response is considered critical in the pathogenesis of AD due to the duality of their functions in the pathophysiology of the disease. Microglia interact with Aβ and tau peptides through the colony-stimulating factor-1 receptor (CSF1R), the toll-like receptor (TLR), and the receptor expressed on myeloid cells 2 (TREM2). In vitro studies have shown that Aβ_1–42_ activates microglia via CD36 and the TLR2-TLR6 heterodimer, leading to the release of proinflammatory factors including IL-1, TNF-α, MIP-1, and MCP-1, promoting secondary immune response activation [141]. Normally, microglial activation promotes phagocytosis and helps clear toxic aggregates. However, the constant and prolonged activation of these cells leads to the continuous release of proinflammatory cytokines, which in turn promote the activation of an inflammatory cascade that generates damage, culminating in neuronal death [137,142,143].

Recent studies have shown that altered microglial activation promotes tau hyperphosphorylation and aggregation in neurons [144]. Microglial cells, in turn, are affected by tau pathology through the degeneration of axons and dendrites and the release of tau aggregates into the extracellular space and are actively involved in the spread of tau pathology to various brain regions [145,146,147,148]. Tau phagocytosis by microglia has been observed to be mediated by the CX3CR1 receptor, which binds directly to tau and promotes its internalization [149,150,151]. Finally, neuroinflammation exacerbates tau and Aβ pathology by increasing cell damage. This leads to neuronal death and consequently to the cognitive impairment observed in AD patients.

### 2.5. Cholinergic Theory in Alzheimer’s Disease

Since the 1970s and 1980s, there have been reports of the importance of acetylcholine (ACh) in cognitive function. ACh is synthesized in the cytoplasm of cholinergic neurons from choline and acetyl-coenzyme A by acetyltransferase (ChAT) and subsequently transported to synaptic vesicles by a vesicular acetylcholine transporter (VAChT). Acetylcholinesterase (AChE) is the enzyme responsible for the degradation of ACh, which is located in the synaptic cleft so that it can be recycled and reused by the presynaptic terminal [152,153]. ACh is a neurotransmitter widely used by the cholinergic system to maintain learning and memory processes, as well as attention, sensory information, and other critical functions. It has been observed that in patients with AD, the loss of synapses is closely related to the loss of cholinergic neurotransmission, which in turn correlates with the low levels of ACh observed in Alzheimer’s patients [19,152,154,155].

Studies in animal models of AD have shown that neurodegenerative processes, neuronal loss, and selective fiber damage in cortical areas innervated by the cholinergic circuit coincide with the formation of Aβ deposits [156,157,158,159,160,161]. In fact, it has been proposed that the cholinergic hypothesis is closely related to the early, asymptomatic stages of AD, since in these stages a presynaptic dysfunction of cholinergic neurons has been observed in the basal nucleus of Meynert, which receives limbic afferents and projects fibers to other cortical regions of the brain [17,18,153]. Given the loss of cholinergic function due to low ACh levels, most therapeutic strategies have focused on AChE inhibition (see below).

## 3. Approved and Conventional Drugs Used to Treat Alzheimer’s Disease

Although the disease was described more than 120 years ago, the current treatment remains unsatisfactory. At present, the Food and Drug Administration (FDA) has approved several different types of drugs, including: (1) acetylcholinesterase inhibitors (AChEIs), and (2) N-methyl-D-aspartate (NMDA) antagonists.

### 3.1. Acetylcholinesterase Inhibitors (AChEIs)

For decades, the main therapeutic approach for patients with AD has focused on the use of inhibitors of AChE, the enzyme responsible for breaking down ACh [19,162]. Cholinergic signaling pathways play a fundamental role in learning and memory processes, and it has been observed that AD patients have a loss of cholinergic innervation in the cerebral cortex [18]. In other words, the severity of the cholinergic deficit closely correlates with the cognitive impairment observed in these patients [19,162]. These drugs act by inhibiting the degradation of ACh in the synaptic cleft, extending its action on its receptors [39,162,163].

The most commonly prescribed inhibitors are rivastigmine [19,164,165], donepezil [166,167,168,169], and galantamine (Figure 2) [170,171,172]. In addition to stabilizing or reducing the progression of cognitive impairment and the behavioral changes observed in patients, results from several studies have shown that these drugs have an important effect in improving cognitive aspects and may delay brain atrophy [19,163,168,169,173,174,175]. However, the side effects associated with these drugs remain a significant challenge, the most common being gastrointestinal events associated with high drug levels [18,120,162,176].

Recently, Çakmak et al. designed and synthesized for the first time six new nicotinic hydrazide derivatives (7–12). Their inhibitory profiles against AChE and BChE were evaluated. The study revealed that hydrazone derivatives containing a dimethylamine moiety, such as rivastigmine, which is used in the treatment of AD, showed remarkable inhibitory activity against the target enzymes even at nanomolar concentrations [177].

From these new hydrazone derivatives, compound 12 exhibited an excellent inhibitory effect and showed the highest potency against both AChE and BChE, surpassing the potency of neostigmine. This AChE inhibitory effect of compound 12 was more than six times higher than that of the reference drug, underscoring the potential therapeutic importance of this compound. In addition, all hydrazone derivatives were found to be more potent AChE inhibitors than the standard molecule, rivastigmine [177].

### 3.2. N-Methyl-D-Aspartate (NMDA) Antagonists

Glutamate plays a fundamental role in several metabolic pathways and is the major excitatory neurotransmitter of the CNS. Glutamate homeostasis, together with ion homeostasis, maintains glutamatergic functions such as neuronal plasticity, synapse formation and signaling, neurotransmission, learning, and memory. Neuron–astrocyte interaction, which regulates the physiological concentration of glutamate in the extracellular space, preserves and maintains these functions. Under pathological conditions, increased glutamatergic neurotransmission caused by elevated glutamate overactivates N-methyl-D-aspartate receptors (NMDAR), leading to excitotoxicity [178,179,180]. This overactivation may even be mediated by the accumulation of Aβ peptides, which would cause increased cytosolic calcium, leading to mitochondrial dysfunction, increased reactive oxygen species (ROS) production, oxidative stress, and, ultimately, cell death [179,180,181]. Overactivation of NMDARs, particularly in the late phase of AD, has been reported to cause neurotoxicity [182,183].

These findings led to the idea that NMDAR antagonists might be a promising therapeutic target for treating AD [178,179,180,183]. That is, by regulating NMDA receptors using competitive and non-competitive antagonists to modulate the damage caused by excess glutamate. Memantine is the main NMDAR antagonist used for the treatment of AD.

Several studies have shown that memantine is effective and safe in animal models, which has encouraged the development of several clinical trials [181,184,185,186]. Memantine is a non-competitive, moderate-affinity NMDAR antagonist that has been extensively studied and used in the treatment of moderate to severe AD as monotherapy or in combination with cholinesterase inhibitors [187]. Most clinical trials have shown that memantine improves autonomy in patients with moderate to severe AD. It increases the likelihood that patients will remain autonomous and thus delays the transition to the dependent stage [187,188,189].

Despite promising results suggesting that N-methyl-d-aspartate (NMDA) receptor antagonists may have a beneficial effect on cognition in patients with AD, the results are still inconsistent.

Conventional drugs are unable to halt the progression of neurodegenerative disease. Therefore, the development of new strategies to detect the disease in its early stages and to improve effective therapies remains a challenge for modern medicine.

## 4. Alternative Approaches for Developing Novel Therapies for Alzheimer’s Disease

### 4.1. Immunotherapy Targeting Aβ

Since Aβ aggregation is the primary mechanism responsible for the neurodegenerative process observed in AD, anti-amyloid therapy has emerged as a promising strategy for treating the disease. A number of monoclonal antibodies targeting different types of Aβ are currently in clinical trials, although three of them have been discontinued due to failure to meet their primary endpoint or toxicity (Table 2).

#### 4.1.1. Active Immunotherapy Focused on Aβ

The most advanced anti-Aβ immunotherapies are vaccines (active immunotherapy) and antibodies (passive immunotherapy). Vaccines involve administering Aβ fragments or the full peptide to induce an immune response to produce endogenous antibodies against Aβ. The use of these therapies has demonstrated their high efficacy in reducing amyloid levels, not only in the brains of transgenic animals (PDAPP mice), but also in clinical trials in patients with moderate to severe AD [190,207,208]. However, although immunization with Aβ resulted in a decrease in Aβ deposits in patients with AD, this reduction does not prevent the progression of the neurodegenerative process [209]. Furthermore, these trials were suspended due to reports of meningoencephalitis in certain patients [208,210].

The first vaccine to be developed and clinically tested against amyloid peptide 1–42 was AN1792, and although, as mentioned above, this vaccine reduced Aβ levels and showed positive effects on neuropsychological tests, 6% of patients developed meningoencephalitis, leading to the termination of the clinical trials [190,208,209,210,211].

CAD106 is another anti-Aβ vaccine, designed to induce N-terminal Aβ-specific antibodies without an Aβ-specific T-cell response [193]. Results from double-blind clinical trials showed that administration of CAD106 or placebo was associated with acceptable safety and tolerability. Among patients reporting serious adverse events, none were considered study drug-related. No clinical or subclinical cases of meningoencephalitis were reported. Although patients develop sufficient anti-Aβ antibodies, the treatment does not reduce Aβ or tau levels [192,193]. More recently, CAD106 has been used in combination with the BACE1 inhibitor umibecestat in people at high risk of developing late-onset AD (homozygotes for the *e4 allele* of *ApoE*). CAD106 administration induced measurable serum Aβ IgG Aβ titers and slower rates of Aβ plaque accumulation, suggesting its use in prevention trials [191].

In 2017, a novel synthetic Aβ peptide vaccine (UB-311) was designed and evaluated in patients with mild to moderate AD. Patients were immunized with three doses of UB-311 at week 0, week 4, and week 12 and were followed up until week 48. Several cognitive test batteries were used to assess efficacy, including the Cognitive Subscale (ADAS-Cog), the Mini-Mental State Examination (MMSE), and the Alzheimer’s Disease Cooperative Study-Clinician’s Global Impression of Change (ADCS-CGIC). The vaccine improved cognition in patients with mild AD, suggesting its use in the early stages is suitable [194,195].

Another vaccine directed against the C-terminus of Aβ-40 (ABvac40) was evaluated for safety and tolerability in a phase I clinical trial in patients with mild to moderate AD. The vaccine was well tolerated and safe, with no cases of edema or microbleeds. It also showed a consistent immune response [196].

#### 4.1.2. Passive Immunotherapy Focused on Aβ

Recently, the most studied antibodies differ in their selectivity for polymorphic variants and recognize epitopes based on the specific portion and conformation of Aβ. For example, aducanumab has a greater affinity for high molecular weight species, solanezumab and ponezumab can bind to the monomeric peptide, lecanemab preferentially targets protofibrils, gantenerumab binds to oligomeric and fibrillar forms, bapineuzumab and crenezumab have been reported to bind to both monomeric and aggregated Aβ, and donanemab targets pyroglutamate Aβ, present only in plaques. It has been suggested that the inconsistent clinical outcomes of this class of drug candidates may be due to this wide range of selectivity [98,203,212,213].

Some monoclonal antibodies such as aducanumab, lecanemab, gantenerumab, bapineuzumab, solanezumab, and crenezumab have been studied and have shown some advantages in animal models as well as in clinical trials. Among these advantages, it is worth mentioning that they can penetrate the blood–brain barrier and the brain parenchyma, inhibit the oligomerization of Aβ and promote its clearance by activating microglial cells and subsequently its degradation through the endosomal/lysosomal system [202,212,214]. However, some of the most common adverse effects observed with the use of Aβ antibodies are amyloid-related imaging abnormalities (ARIA). These are characterized by cerebral microbleeds/hemosiderosis (ARIA-H) and edema (ARIA-E). The mechanism of ARIA-E is not completely understood. However, it has been proposed to involve direct binding of Aβ to cerebral amyloid angiopathy or to accelerate its formation. In addition, some antibodies directed against Aβ (aducanumab, bapineuzumab, donanemab, and lecanemab) have been shown to induce accelerated brain volume loss in Alzheimer’s patients, with a major impact on ventricular enlargement [202,212,214].

Research into immunotherapy for AD is still ongoing, with developers exploring subcutaneous administration to eliminate the need for infusion, which would make home administration more feasible [212].

In this section, we focus on aducanumab and lecanemab, the only antibody therapies approved by the Food and Drug Administration (FDA) as disease-modifying drugs for the treatment of Alzheimer’s disease.

##### Aducanumab

Aducanumab, which received accelerated approval from the FDA in June 2021 after two phase III clinical trials (EMERGE and ENGAGE), is a recombinant human antibody developed by Biogen. It was derived from a blood lymphocyte library collected from healthy elderly donors with no evidence of cognitive impairment or unusually slow cognitive decline. The antibody is indicated for people with early AD or mild cognitive impairment (MCI) with a positive brain amyloid PET scan [201,212].

Aducanumab is administered by intravenous infusion at 10 mg/kg every four weeks; however, the phase I clinical trial demonstrated that a dose ≤ 30 mg/kg was generally well tolerated without serious adverse events. Since the phase I clinical trial, aducanumab has demonstrated dose- and time-dependent reductions in brain Aβ plaques, as well as slowing of decline on the CDR-SB, MMSE, ADAS-Cog 13, and Alzheimer’s Disease Cooperative Study-Activities of Daily Living for Mild Cognitive Impairment (ADCS-ADL-MCI). Some biomarkers of AD such as tau PET, CSF p-tau, and plasma p-tau 181 were reduced after treatment with aducanumab [202,212].

The most common adverse events observed with the use of aducanumab were ARIA-E (35%) and ARIA-H (19.1%). The main risk factors identified for the development of ARIA-E were antibody dose and the presence of the apolipoprotein E4 (*ApoE4*) allele [202,212].

Using inhibition ELISA, immunodepletion, and surface plasmon resonance (SPR) with synthetic Aβ and brain extracts from Alzheimer’s patients, Söderberg et al. [202] showed that aducanumab does not bind to oligomers, binds weakly to monomers (7300 ± 990 nM) and instead, it preferentially binds to protofibrils of Aβ with high affinity (2.2 ± 1.0 nM for small and 0.79 ± 0.10 nM for large protofibrils), and that this binding is driven by a very fast apparent association rate of 2.5 ± 0.53 × 10^7^ M^−1^ s^−1^ for small and 3.8 ± 0.56 × 10^7^ M^−1^ s^−1^ for large protofibrils, and an apparent dissociation rate of 5.2 ± 1.7 × 10^−2^ s^−1^ and 3.0 ± 0.56 × 10^−2^ s^−1^, respectively, for the small and large fibrils. These results support the study by Arndt et al. in 2018, who used negative staining electron microscopy to image A_β1–40_ and Aβ_1–42_ fibrils incubated with an analog of aducanumab conjugated to 10 nm gold particles and found that Aβ fibrils incubated in this way were decorated with a much higher number of gold particles than seen in the background [203]. In the same study, the authors showed that the epitope for aducanumab is conformed by residues Glu3 to Asp7 of the Aβ peptide, with residues Phe4 and His6 forming the core epitope. To further understand the molecular basis of antigen recognition, they determined the structure of the Fab of aducanumab complexed with the A_β1–11_ peptide at 2.4 Å resolution. They found that the Fab fragment of aducanumab binds to the N-terminus of Aβ in an extended conformation, which is different from that seen with other antibodies. Furthermore, the interaction between aducanumab and Aβ is very weak compared to other Aβ antibodies [203].

The mechanism of action of aducanumab has been progressively elucidated, with Linse et al. in 2020, showing that it reduces the secondary nucleation rate of A_β1–42_, decreasing the effective rate constant for this process by approximately 40% at concentrations ranging from 250 pM to 100 nM [201]. However, no changes in the growth of preexisting aggregates were observed. It has also been observed that this reduction in aggregation is accompanied by a decrease in the production of oligomers. These data confirmed that aducanumab predominantly interacted with species that are unique to secondary nucleation. Using microfluidic diffusional sizing, the authors found one molecule of aducanumab per 4.5 ± 0.6 Aβ42 monomers in the fibril, meaning that fibrils can be completely coated with aducanumab along their entire length [201].

Several ways to improve the efficacy of aducanumab have been explored, including opening the blood–brain barrier with ultrasound [215], improving meningeal lymphatic drainage [216], or co-treating the antibody with the chaperone Brichos [201], all of which have shown promising results. However, all of these options are at the preclinical stage and some of them are still being studied in mouse models.

##### Lecanemab

Lecanemab is a humanized IgG1 derived from murine mAb158 that specifically binds to large and soluble aggregates of Aβ (oligomers and protofibrils), which are the most neurotoxic and contribute to the pathogenesis of AD. It received full FDA approval in July 2023 and is indicated for patients with MCI or mild Alzheimer’s dementia who have a positive amyloid PET scan or cerebrospinal fluid findings consistent with AD. It is administered by intravenous infusion every two weeks at a dose of 10 mg/kg [98,212].

In a phase I clinical trial, lecanemab was shown to have a serum half-life of 7 days. The phase II clinical trial reached its endpoint (change from baseline on the ADCOMS) at 12 months, demonstrating that the dosing arm had an 80% probability of slowing decline on the ADCOMS by 25% more than the placebo; this phase also showed that APOE ε4 carriers are at greatest risk for ARIA. Lecanemab reduced amyloid PET plaque levels (−55.48 centiloid change) compared to placebo (+3.64 centiloid change) in the CLARITY AD phase III clinical trial. In addition, all CSF and plasma biomarkers except neurofilament light favored lecanemab over placebo. The most common adverse events were infusion-related reactions (26.4%), ARIA-H (17.3%), and ARIA-E (12.6%) [98,151,205,212,217] {Cummings, 2024 #4;Fedele, 2023 #5;Shi, 2022 #5;Swanson, 2021 #6;van Dyck, 2023 #7;Fedele, 2023 #1;Shi, 2022 #3;Swanson, 2022 #5;van Dyck, 2023 #4}.

Lecanemab binds weakly to Aβ monomers (≥25,000 nM) and instead binds to both small and large Aβ protofibrils with an IC50 of 0.8 nM. Söderberg et al. performed an immunodepletion assay to evaluate the interaction of lecanemab with Aβ protofibrils. In this assay, serially diluted lecanemab was allowed to interact with synthetic Aβ protofibrils, and the antibody–Aβ complex was depleted using magnetic protein A beads. Near complete depletion of the protofibrils was observed at a lecanemab concentration of 10 ng/mL; the EC50 for depletion of the protofibrils indicated a slight preference for the large protofibrils over the small (3.5 and 5.3 pM, respectively). The authors also calculated the avidity of lecanemeb for binding to small and large protofibrils using SPR and determined a value of 0.97 ± 0.66 and 0.16 ± 0.07 nM, respectively. The increased affinity was mainly determined by the dissociation rate, which was approximately three orders of magnitude slower than binding to the monomer, 4.5 ± 1.7 × 10^−4^ s^−1^ and 1.1 ± 0.36 × 10^−4^ s^−1^, for the small and large protofibrils, respectively. The apparent affinity of the antibody for Aβ fibrils was 1.8 ± 0.93 nM, approximately 2-fold and 11-fold lower than for small and large protofibrils, respectively, indicating that lecanemab prefers to bind to protofibrils rather than fibrils [202].

### 4.2. Immunotherapy Targeting Tau

Tau immunotherapy has progressed from proof-of-concept studies to more than a dozen clinical trials in AD. However, there are still some obstacles to overcome in this area of research. For example, it is difficult to assess which strategy would produce better results, i.e., whether immunotherapy should target abnormal tau phosphorylation, inhibit aggregation and paired helical filament formation, or promote clearance of neurofibrillary tangles [218,219,220].

Some of the antibodies tested in recent years can prevent tau-induced neurotoxicity and tau seeding. Many of them have been detected inside neurons (in the endosomal/lysosomal pathway), facilitating clearance of the protein. In addition, these antibodies have also been detected in the cytosol. They bind to the Fc receptor TRIM21, which promotes tau ubiquitination and proteasomal clearance. In contrast, antibodies that bind to extracellular tau promote phagocytosis by microglial cells [218].

A variety of tau epitopes have also been studied in preclinical and clinical studies, including unphosphorylated, phosphorylated, oligomeric, conformational, and truncated tau. Phospho-serine 396,404 has been one of the most studied. However, more tau antibodies in clinical trials target the N-terminus [218,220].

The isotype of the antibody is also controversial, with human IgG1 being the most effective in promoting microglial phagocytosis and IgG4 being the least effective. IgG1 and IgG4 tau antibodies are currently in clinical trials [220] (Table 3).

#### 4.2.1. Active Immunotherapy Focused on Tau

The first vaccine developed for tau pathology consisted of active immunization with a phosphorylated tau epitope in P301L tangle model mice. The authors observed that tau aggregates in the brain were reduced, and the progression of tangle formation was slowed [235].

Currently, clinical trials with two vaccines have shown promising results. Novak et al. have reported a phase I/II, randomized, double-blind, placebo-controlled study of AADvac1 [221]. This vaccine was designed against N-terminally truncated tau fragments (amino acids 294–305 of tau protein). The results showed that AADvac1 is safe and immunogenic [221]. Subsequent results have shown that AADvac1 tau immunotherapy can reduce plasma biomarkers of neurodegeneration and neuroinflammation. In addition, AADvac1 non-significantly reduced the loss of cerebral cortical tissue in several regions [222].

Another active antibody used is ACI-35. This liposome-based vaccine is directed against phosphorylated residues pSer396,404. Preclinical trials on P301L tauopathy mice demonstrated its efficacy. Its safety and efficacy were subsequently tested in a group of patients with mild to moderate AD. The results of the clinical trial demonstrated that the vaccine was safe and well-tolerated [223].

#### 4.2.2. Passive Immunotherapy Focused on Tau

Passive immunotherapy has received increased attention, possibly due to its greater safety, as the presence of adverse effects in patients is reduced. E2814 is a high-affinity humanized IgG1 antibody that recognizes the repeat domains of tau, both the 3R and 4R forms; it is capable of inhibiting tau aggregation, as well as seeding in vitro and in vivo [224]. The antibody safety, tolerability, and immunogenicity were tested in healthy subjects, with no adverse effects reported [236]. A phase Ib/II trial with mild-to-moderate cognitive impairment is under development.

Another antibody employed is JNJ-63733657, which is a humanized IgG1/kappa monoclonal anti-phospho tau antibody with a high affinity for p-tau217. Preclinical studies have demonstrated that JNJ-63733657 is capable of reducing the pathological spread of tau. Phase I randomized, double-blind, placebo-controlled trials were conducted in healthy subjects and patients with prodromal or mild AD, demonstrating high tolerability in both groups. Furthermore, a reduction in p217+tau in the cerebrospinal fluid was observed following the administration of JNJ-63733657, indicating that it could be an effective strategy for slowing the progression of AD. A phase II trial is being developed to evaluate JNJ-63733657 in patients with early-stage [225,226].

Bepranemab is a humanized, monoclonal IgG4 antibody that binds to the central region of tau (amino acids 235–250). The initial trial in healthy subjects indicated the absence of drug-related adverse events or alterations in safety outcomes. Phase I trials in subjects with progressive supranuclear palsy demonstrated the absence of adverse events following treatment. The phase 2 trial aims to randomly evaluate 450 individuals with mild cognitive impairment or mild AD dementia. This group of patients will receive one to two doses of bepranemab or placebo for 80 weeks, with results anticipated in 2025 [227,228].

PRX005 is an IgG1 antibody that targets the microtubule-binding region of the 3R and 4R isoforms. It is capable of recognizing both phosphorylated and non-phosphorylated forms of tau, as well as NFTs, in the brain tissue of patients with AD. In animal models of tauopathy and amyloidosis, this antibody has been demonstrated to inhibit tau aggregation and p-tau accumulation. A phase I study is currently underway with the objective of evaluating the safety and tolerability of PRX005. The study, which was conducted in healthy subjects, demonstrated that the antibody is safe for use [229].

Lu AF87908 is an IgG1 antibody that has been raised against p-tau396/404. It has been demonstrated to exhibit high binding specificity to tau in post-mortem brain tissue from individuals who have been diagnosed with AD or other tauopathies (hC10.2) [230]. In animal models (C10.2), the antibody was also demonstrated to possess the capacity to diminish tau seeding in vitro and in Tg4510 mice, as well as in microglia cultures, where it facilitated tau uptake and lysosomal-mediated degradation [231,232]. The ongoing phase I trials (NCT04149860) have demonstrated that the antibody is well tolerated and safe in both healthy individuals and patients with AD.

PNT001 is a humanized monoclonal antibody that exhibits specificity for the cis-conformer of the phosphorylated tau protein at threonine 231 (cis-pT231 tau). Preclinical trials have demonstrated that treatment with PNT001 results in the elimination of pathological tau from the brain and an improvement in both neuronal degeneration and cognitive decline in a mouse model [234,237]. Phase II trials have evaluated the safety and tolerability of this antibody in healthy subjects and patients with AD [233].

## 5. Potential Medicinal Plants for Alzheimer’s Disease

The use of medicinal plants has been widely explored. The extracts obtained from them contain compounds such as phenols, flavonoids, terpenoids, and alkaloids. They have long been considered as an alternative for the treatment of various neurological diseases, such as AD, Parkinson’s disease, Huntington’s disease, multiple sclerosis, etc. A wide variety of plants have been studied for their therapeutic potential in the treatment of AD [238,239,240,241,242,243,244]. In this section, we focus on analyzing the studies carried out on *Curcuma longa* and *Ginkgo biloba*.

### 5.1. Curcumin (Curcuma longa)

Curcumin is a polyphenolic compound derived from the rhizomes of *Curcuma longa*. It has long been used in traditional Ayurvedic medicine to treat various conditions such as obesity, rheumatism, cancer, and respiratory infections. Its chemical structure consists of a chain of unsaturated aliphatic and aromatic groups, including two beta-keto, carbon–carbon double chains, and 3-methoxy-4-hydroxybenzene [245]. Because of its pleiotropic effects, which include anti-amyloidogenic, anti-inflammatory, antioxidant, and metal-chelating properties, it has been proposed as a potential treatment for AD.

The anti-amyloidogenic properties of curcumin have been extensively studied in vitro [246] and in vivo models for over two decades. Yang et al. [247] found that curcumin supplementation in the diet or peripherally administered to aged Tg2576 mice could bind and prevent the formation of Aβ oligomers and fibrils. This observation was confirmed by Garcia-Alloza et al. [248], who showed that curcumin, after systemic administration to transgenic mice, crossed the blood–brain barrier and bound to Aβ deposits, preventing their aggregation (with an IC50 of about 0.81 μM) and degrading pre-aggregated Aβ (with an IC50 of 1 μM) [248]. In addition, they demonstrated that curcumin promoted a reduction in plaque size and number, as well as accelerated clearance of existing plaques and prevention of new plaque formation. These results are consistent with Zheng et al. [245] who reported that the reduction of Aβ by curcumin treatment was due to inhibition of BACE1.

Curcumin and some of its derivatives are known to reduce microglial and astrocyte activation, as well as proinflammatory cytokine synthesis, and promote the anti-inflammatory M2 phenotype [246,249,250,251,252]. The mechanisms by which curcumin reduces inflammatory responses in the central nervous system are not fully understood. Several studies in AD and other pathologies suggest that this is achieved by inhibiting TLR4/MyD88/NF-κB signaling in microglial cells and macrophages [253]. Similarly, curcumin decreased TLR4 and NF-κB hyperactivity by increasing TREM2, a negative regulator of TLR4, in LPS-activated BV2 cells [250]. Furthermore, Shi et al. demonstrated that curcumin can block the ERK1/2-p38 signaling pathway on Aβ-activated microglia [252].

Another mechanism by which Aβ induces neuronal death is through increased ROS, decreased expression of antioxidant enzymes, decreased base excision repair (BER)-related proteins, and decreased antioxidant response element pathways [254]. Mishra et al. demonstrated that when primary rat hippocampal neuronal cultures exposed to oligomeric Aβ are pretreated with tetrahydro curcumin, a curcumin metabolite, a reduction in ROS levels is observed [255]. Similarly, Kumar et al. found that curcumin administered to aluminum chloride-treated rats prevented the increase in malondialdehyde (MDA), nitrite levels, and depletion of reduced glutathione. It also significantly increased the levels of antioxidant enzymes such as superoxide dismutase (SOD), catalase (CAT), and glutathione-S-transferase [256].

### 5.2. EGb 761 (Ginkgo biloba)

*Ginkgo biloba* is a tree native to China; its leaves are commonly used to make an extract (EGb761) that is believed to alleviate cognitive disorders. This extract is composed of 6% of terpenoids (ginkgolides A, B, C, J, M, and bilobalide), 24% flavonoid glycosides (quercitin, kaempferol, isorhamnetin, etc.), and 5–10% organic acids. The exact mechanism by which EGb761 alleviates neurological symptoms remains undiscovered, but it has been shown that its activity is determined by the properties of each of its components, with flavonoids and terpenoids being the pharmacologically relevant ones [257].

The antioxidant activity attributed to the flavonoid fraction is the most studied property of *G. biloba* extract [257,258,259,260]. In N2a cells transfected with the Swedish APP mutation and treated with EGb761, the levels of MDA and ROS were significantly decreased, whereas those of antioxidant enzymes such as SOD, total CAT, and glutathione peroxidase were increased. Similarly, Aβ levels were reduced [261]. These results are consistent with those of Shi et al. in 2009 [259], who demonstrated that quercitin and ginkgolide B can block early Aβ-induced signaling, reduce the generation of ROS, partially prevent mitochondrial dysfunction, attenuate the activation of caspase 3 and JNK, and completely reverse the activation of ERK1/2 and AKT. In addition, EGb761 can reduce glutamate excitotoxicity by modulating redoxosome-p66c signaling in SH-SY5Y cells [262].

Protection of the cholinergic system is one of the mechanisms involved in the improvement of memory and learning observed in humans consuming *G. biloba* extract. In 2019, Zhang et al. found that rats with cognitive dysfunction induced by scopolamine administration and treated with EGb761 showed a significant reduction in AChE activity and improved Ach levels compared to control rats [263]. They proposed that this effect is mediated by the upregulation of the BDNF-TrKB signaling pathway. In AD, inactivation of this pathway is associated with synaptic damage. In 2018, Kuo et al. showed that Ginkgolide A, a pure compound extracted from *Ginkgo biloba,* attenuated Aβ-induced abnormal depolarization in primary cortical neurons of mice [264]. Similarly, ginkgolide B has been shown to reduce neuroinflammation induced by oligomers of Aβ_1–42_ by decreasing the levels of the NLRP3 inflammasome, a multimolecular complex present in microglia and implicated in the pathogenesis of AD. In addition, ginkgolide B promotes the polarization of microglia from the pro-inflammatory M1 phenotype to the anti-inflammatory M2 phenotype [114]. Bilobalide also acts on astrocytes to inhibit the STAT3-dependent expression of TNF-α, IL-1β, and IL-6, and to promote the expression of neprilysin, insulin degradation enzyme, and matrix metallopeptidase-2, all of which are enzymes that help degrade Aβ [265]. Bilobalide treatment increased lincRNA-p21 levels, which suppressed STAT3 signaling and improved learning and memory in APP/PS1 AD mice [266]. Finally, other studies have shown that bilobalide promotes neurogenesis and synaptogenesis in hippocampal neurons. These processes may be mediated by cyclic AMP response element binding protein [267].

## 6. Modifying the Gut Microbiota to Treat Alzheimer’s Disease

### 6.1. Probiotics

A promising therapeutic approach to prevent or slow Alzheimer’s disease is the homeostatic restoration of the gut microbiota (microbiotherapy). The goal of therapeutic strategies is to reverse dysbiosis by restoring a healthy balance of gut bacteria. This can improve the integrity of the intestinal barrier, reducing pro-inflammatory processes, and the translocation of pathophysiological neuroactive metabolites. In addition, restoring microbiota homeostasis has beneficial effects on cognitive function by promoting the production of neurotransmitters and beneficial metabolites, such as short-chain fatty acids, which have anti-inflammatory and neuroprotective properties [268].

Probiotic supplementation has been proposed as a therapeutic strategy to regulate the microbiota in AD. Probiotics are live microorganisms, including bacteria similar to those naturally found in the human gut. Probiotics can improve the balance of the microbiota and promote better gut and brain function when taken in sufficient amounts [268].

In an uncontrolled clinical trial, patients with AD who received supplementation with fermented milk containing the probiotics *Acetobacter aceti*, *Acetobacter* sp., *Lactobacillus delbrueckii delbrueckii*, *Lactobacillus fermentum*, *Lactobacillus fructivorans*, *Enterococcus faecium*, *Leuconostoc* spp., *Lactobacillus kefiranofaciens*, *Candida famata*, and *Candida krusei* (2 mL/kg/day) for 90 days showed improvement in memory, visuospatial/abstraction, and executive/language functions. They also showed a decrease in inflammatory markers, oxidative stress, and cellular damage in the blood [269].

In another randomized, double-blind, controlled clinical trial, co-supplementation of the probiotics *Lactobacillus acidophilus*, *Bifidobacterium bifidum*, and *Bifidobacterium longum* plus selenium for 12 weeks improved cognitive function in patients with neurodegenerative diseases. They also experienced increased antioxidant markers and decreased serum high-sensitivity C-reactive protein (hs-CRP), insulin, triglycerides, very low-density lipoprotein (VLDL), and total/HDL cholesterol [270].

### 6.2. Fecal Microbiota Transplantation

Fecal microbiota transplantation (FMT) is a procedure in which microbiota are harvested from the feces of a healthy donor and transferred into the gastrointestinal tract of an affected recipient. The goal of FMT is to restore a healthy balance of bacteria in the Alzheimer’s recipient’s gut to potentially improve mood, memory, and cognition. This strategy has previously been used in the treatment of recurrent gastrointestinal infections caused by *Clostridium difficile*, a bacterium that can cause diarrhea and severe digestive problems, particularly in cases where antibiotic resistance has been observed [271,272].

A case report describes a patient with AD and recurrent *Clostridioides difficile* infection (CDI) who was treated with a single infused dose of FMT (300 mL). The patient’s CDI symptoms resolved, and he also showed cognitive improvements as assessed by the MMSE at 2 and 6 months after FMT [273]. In another similar report, an AD patient with CDI was treated with two doses of FMT administered by colonoscopy three months apart. The patient showed improvement in cognition and mood after FMT. Analysis of her gut microbiota revealed enrichment of the genera Bacteroidales and Bacteroidia, which are associated with cognitive function, was found in the analysis of her gut microbiota. In addition, changes in short-chain fatty acid indicators were found after FMT, which may be related to functional improvement [274].

Cheng et al. (2023) also performed FMT by capsule administration (1 g of fecal mixture per capsule) in an open-label, single-arm clinical trial [275]. AD patients took three doses of 40 capsules every two weeks. Six months after FMT, there was no improvement in cognitive function. However, there was no deterioration either. FMT altered the composition of the gut microbiota, with individuals experiencing an increase in *o_Fusobacteriales*, *f_Fusobacteriaceae*, *c_Fusobacteria*, *p_Fusobacteria*, *f_Eggerthellaceae*, and *g_Prevotella_7*, and a decrease in *s_un_g_Lachnospira* and *g_Lachnospira*. In addition, metabolomic analysis revealed that FMT led to a decrease in bilirubin, 4-hydroxypheoxyacetate, phloracetophenone, 4-hydroxy-5-(3,4,5-trihydroxyphenyl)pentanoic acid, alpha-furyl methyl diketone, Xi-2,3-dihydro-3,5-dihydroxy-6-methyl-4H-pyran-4-one, 6-(Hydroxymethyl)-7-methoxy-2H-chromen-2-one, alpha-triticene, squamostanal A, while there was an increase in 3b,12a-dihydroxy-5a-cholan-24-oic acid, 25-acetylvulgareoside, deoxycholic acid, 2(R)-hydroxydocosanoic acid, and p-anisic acid [275].

### 6.3. Antibiotics

The use of antibiotics as a therapeutic strategy to modify gut microbiota to improve cognition in AD patients is controversial. This is because antibiotics, especially broad-spectrum antibiotics, can affect both pathogenic and beneficial bacteria. As a result, antibiotics disrupt the composition of the gut microbiota, reduce its biodiversity, and delay recolonization for a long time after administration. This unintended effect results in antibiotics causing digestive problems such as diarrhea, bloating, and food sensitivity, as well as potential changes in the permeability of the intestinal barrier with consequent effects on brain function. In addition, prolonged or inappropriate use of antibiotics can promote the development of bacteria that are resistant to these drugs. These bacteria can spread in the gut microbiota and transfer their resistance genes to other bacteria, making future infections more difficult to treat [276,277].

Despite the lack of results and the controversy surrounding the use of antibiotics, it remains an alternative under investigation. This is because there is growing evidence that AD may have an infectious origin. This hypothesis proposes that increased Aβ peptide may be a mechanism of antimicrobial protection induced by the innate immune system against an invasion of the brain by pathogenic microorganisms such as bacteria and viruses, including those that enter via the intestinal route. Early treatment with antibiotics, by controlling the infection, would mean a reduction in Aβ peptide, prevention of its accumulation, and thus the proper execution of cognitive-behavioral functions [278].

A study was conducted in patients with AD and gastrointestinal infection with *Helicobacter pylori* (Hp). Antibiotics were administered to treat the bacterial infection. In addition, all patients received the same ChEI during the 2-year follow-up period of the study. After this period, only patients in whom Hp eradication was achieved showed improvement in cognitive function, suggesting that treatment may have altered the progressive nature of the disease [279]. Whether there were beneficial changes in the gut microbiota after Hp eradication that could potentially benefit brain function remains to be determined.

## 7. Emerging Therapeutic Strategies

### 7.1. Metformin

For decades, metformin has been prescribed as the first line of medication for the treatment of type 2 diabetes and metabolic syndrome (MetS), a cluster of metabolic alterations such as dyslipidemias, hyperglycemia, insulin resistance, and obesity. MetS is strongly related to the development of AD since both conditions share underlying molecular and cellular disruptions such as oxidative stress, chronic inflammation, mitochondrial dysfunction, and decreased glucose metabolism, among others. In this sense, AD is identified by many as a new kind of diabetes, type 3. There is a large body of evidence supporting the potential beneficial properties of metformin in people with AD. Preclinical studies have demonstrated that metformin is able to reduce brain gliosis, rescue glucose oxidation, and improvement of cognitive function in animal models of neurodegenerative diseases [280,281,282]. Nonetheless, there is controversial data pointing out that in long-term treatment or under certain metabolic circumstances, metformin may cause deleterious effects on cognition [283], so the translation to clinical use may be taken cautiously.

### 7.2. microRNAs

Since their discovery in the early 1990s [284,285,286], microRNAs (miRNAs) have been investigated given their therapeutic potential in various human diseases [287,288,289,290,291,292] including COVID-19 [293,294,295,296]. miRNAs are a class of short (~18–24 nucleotides) non-coding RNAs that inhibit gene expression by acting as post-transcriptional regulators [297]. Inhibition of gene expression by miRNAs can occur in several ways: by inhibiting translation initiation or by stimulating mRNA decay [298,299]. Consequently, miRNAs are involved in the regulation of many cellular functions as cell differentiation, development, and homeostasis.

A multistep sequence is involved in miRNA biogenesis: in the nucleus, they are initially transcribed from DNA sequences into primary miRNA (pri-miRNA), then processed into precursor miRNA (pre-miRNA), and finally, in the cytoplasm, the pre-miRNA is processed into mature miRNA [298,300]. The regulatory mechanisms for the biogenesis and functionality of miRNAs have already been described [298,300], so we will not address this issue in this review.

Later, some studies showed that miRNAs can be secreted into extracellular fluids (within vesicles or associated with protein carriers) such as blood (serum, plasma), urine, cerebrospinal fluid, saliva, milk, lacrimal, seminal fluids, and others [299,300,301,302,303,304]. These circulating miRNAs (c-miRNAs) have been proposed not only as promising biomarkers for various diseases but also as signaling molecules to mediate communication between cells [300,301,304,305].

### 7.3. microRNAs in Alzheimer’s Disease

Because of their properties, miRNAs have been positioned as new players in the therapeutic field of AD.

For example, changes in the expression of specific miRNAs have been found in preclinical and clinical studies, suggesting that they may contribute to the pathophysiology of AD [306,307,308,309,310,311]. However, the molecular mechanisms involved in these actions are still under investigation.

In this regard, miR-485-3p has been associated with Aβ plaque formation, tau pathology development, upregulation of inflammatory response, and cognitive decline in AD [312]. miR-485-3p was overexpressed in brain tissue and cerebrospinal fluid (CSF) of AD patients. More importantly, miR-485-3p antisense oligonucleotides (ASO; once weekly/2 weeks, intracerebroventricular injection) demonstrated therapeutic potential in vivo in a transgenic mouse model of AD [312]. Furthermore, miR-650 was found to be upregulated in the cortical tissue of AD patients [198]. Its overexpression can inhibit the expression of CDK5, a kinase associated with the pathogenesis of AD [198]. Consequently, overexpression of miR-650 reduced the number of plaques and Aβ levels in APP/PSEN1 mice (AD model) [198]. These results not only highlight a novel miR-650-CDK5 regulatory axis but also suggest a new potential therapeutic target for AD.

Another recent study found that the increased level of miR-22-3p in the hippocampus could improve cognitive abilities and Aβ deposition by targeting a key neuroinflammatory signaling pathway in a mouse model of AD [313].

On the other hand, loss of neurogenesis in AD has been associated with decreased expression of miR-132, both in human tissue and in two amyloidosis mouse models of AD. miR-132 appears to be a key regulator for the induction of neurogenesis in the dentate gyrus in vivo. Restoring miR-132 levels in the adult brain reverted not only neurogenic but also memory deficits in vivo, in mouse models [314].

A study has demonstrated not only biphasic changes in miR-331-3p and miR-9-5p expression, which can dynamically regulate autophagic activity during AD progression but also prevention during the late stage of AD [315]. miR-331-3p and miR-9-5p showed lower levels in the early stage and higher levels in the late stage of AD in a mouse model [315]. Both miRNAs targeted autophagy receptors (Sqstm1/Optn), and their inhibition during the late stage of AD with antagomirs (antisense inhibitor oligonucleotides aimed to reduce the functionally available endogenous miRNA) promotes the autophagic clearance of Aβ by targeting these autophagy receptors [315]. These actions may ameliorate memory loss and improve mobility in vivo. Recently, miR-1273g-3p has been suggested as a biomarker for early diagnosis of AD [316]. miR-1273g-3p is increased in the CSF of early-stage AD patients, which contributes to the formation of Aβ plaques due to its primary interaction with genes associated with oxidative stress induction and mitochondrial dysfunction, in in vitro AD model [316].

Research on miRNAs has included an analysis of both non-circulating (brain tissue) and circulating (blood, plasma, CSF, etc.) miRNA expression profiles. These studies demonstrate the therapeutic potential of miRNAs as biomarkers for AD diagnosis, considering that miRNAs can be dynamically expressed during AD progression [315,316,317]. Also, their efficacy as miRNA-based therapies for AD treatment has been validated at the preclinical level using miRNA mimics (synthetic miRNAs used to restore endogenous miRNA levels) as well as antagomirs [312,314,315]. Although miRNA-based therapeutics for AD are in preclinical testing, their evaluation in clinical trials is still far from being feasible. Currently, only clinical trials with miRNAs for AD diagnostic are being conducted (consult ClinicalTrials.gov).

Finally, there is still a large gap between the development of new technologies focused on targeting the delivery of miRNAs to the brain and their availability as a therapeutic option for AD treatment.

### 7.4. Proteolysis Targeting Chimera (PROTAC)

PROTACs are hybrid complexes composed of two molecules connected by a linker. One of these molecules acts as a ligand for the ubiquitin E3 ligase, while the other specifically binds to the target protein. PROTACs facilitate the interaction between the E3 ligase and the target protein, resulting in its polyubiquitination and subsequent degradation via the ubiquitin–proteasome system. This mechanism has the potential to completely eliminate the target protein, offering a more effective and sustained therapeutic strategy compared to traditional protein inhibition approaches [318,319,320].

In the context of Alzheimer’s disease, small-molecule PROTACs capable of crossing the blood–brain barrier have been designed, targeting hyperphosphorylated Tau protein. In a study, treatment with these compounds significantly reduced Tau levels in cell cultures and in the hippocampus of an animal model of this pathology. Furthermore, in treated mice, the intervention was associated with improvements in cognitive performance [3].

The chimeric nature of PROTACs enables the exploration of diverse molecular combinations, including peptidic PROTACs, which may offer advantages over small-molecule PROTACs due to their ability to mediate protein-protein interactions with higher affinity. For instance, a Tau-targeting peptidic PROTAC has been developed and encapsulated in lipid nanoparticles with high blood–brain barrier penetration efficiency, utilizing DNA intercalation technology. Treatment with this nanocomplex-PROTAC significantly degraded hyperphosphorylated Tau protein in both cell cultures and the hippocampus and cerebral cortex of a transgenic mouse model of Alzheimer’s disease. Additionally, the treatment improved performance in behavioral tests related to learning and memory in these animals [321].

However, a limitation of the therapeutic potential of using peptide PROTACs is their high molecular weight, their low cellular penetration, as well as their poor ability to penetrate the blood–brain barrier (BBB) [322,323]. Thus far, PROTAC technology has been limited to intracellular targets. However, alternative strategies are being developed to expand its application to other therapeutic targets, such as beta-amyloid [324].

## 8. Nanotherapy in Alzheimer’s Disease

As mentioned above, the main drugs currently approved by the FDA for the treatment of Alzheimer’s disease, AchE, and NMDAR inhibitors, are administered orally, so high doses are required for the therapeutic fraction to reach the brain and overcome oral absorption barriers, hepatic metabolism, and finally cross the BBB [325,326,327]. In addition, most of these drugs and their dosage forms cause a high incidence of side effects such as nausea, vomiting, and diarrhea due to their effects on peripheral tissues, which reduces the quality of life of patients. Thus, the search for new methods of drug administration will make it possible to increase the efficacy of existing treatments. The primary goal of drug delivery is to be able to deliver the therapeutic agent to the site of action, minimize the adverse effects of the drug on healthy tissues or organs, and control the release to avoid cyclic overdosing/underdosing [328,329,330].

For this reason, interest in the use of nanotechnology for the treatment of neurodegenerative diseases has increased significantly in recent years. In general, nanoparticles are between 1 and 100 nm in size, allowing them to penetrate biological barriers such as the BBB [29].

### 8.1. Targeting Transport Across the BBB

BBB is a diffusion barrier that prevents harmful substances from entering the brain, maintaining brain homeostasis. Under normal conditions, BBB flux is regulated by different transport mechanisms, known as paracellular and transcellular pathways [331,332]. Ions and small molecules passively cross the BBB through the complex junctions between cells in the paracellular pathway. This pathway depends on the solute’s concentration, charge, and size, as well as gradients. Changes in BBB permeability caused by paracellular transport are governed by adhesion forces generated at the junctions between endothelial cells and contractile forces generated in the endothelial cytoskeleton.

The transcellular pathway transports solutes between cells. This pathway is dynamically regulated during disease development and is essential for transport across the BBB [332]. The transcellular route involves diffusion across luminal and abluminal membranes, transcytosis mediated by receptors and transporters, efflux transport, and endocytosis of positively charged molecules. Lipophilic, non-polar, low-molecular-weight molecules can cross endothelial cell membranes and the BBB via the transcellular pathway. In the diagnosis and treatment of CNS disorders, the BBB plays an important role. Some diseases, including stroke, diabetes, seizures, and AD, can alter it directly [333]. A compromised BBB can lead to altered homeostasis in the brain, resulting in ionic imbalances and the infiltration of immune cells and molecules that can lead to the dysfunction and degradation of neurons. Conventional drug delivery methods are impeded by the BBB, limiting effective drug delivery to the brain. Therefore, several strategies, which can be divided into two broad groups, invasive and non-invasive, have been developed to deliver drugs across the BBB into the CNS [334,335,336,337,338,339,340] (Figure 3).

Since nanoparticle size is in the range of proteins and nucleic acids, they can interact with these biomolecules, amplifying their efficacy. Another property is that biocompatibility is increased by facilitating the transport of therapeutic compounds, which could reduce the dose used in Alzheimer’s patients [359,360]. For example, the use of lipid-soluble nanoparticles targeted to endothelial cells would increase the rate of drug transport by increasing its adsorption through endocytosis or through lipophilic transcellular pathways [361,362]. On the other hand, the development of nanoparticles with specific receptors could also improve drug uptake. Thus, the use of nanotechnology in drug delivery systems improves bioavailability and can help to safely deliver drugs to specific molecular targets, thereby reducing adverse effects [362].

The transport of nanoparticles across the BBB also depends on their physicochemical properties. Nanoparticles have a limited ability to cross the blood–brain barrier on their own and thus gain access to the brain. Therefore, several ways have been explored to enable them to cross the BBB, taking advantage of the various physiological processes for crossing the BBB. Delivery methods that have been investigated include the use of receptor-mediated endocytosis, transcytosis, or transporters, focused ultrasound in synergy with microbubble delivery or by various means of surface modification of nanoparticles and local drug delivery from nanoparticles and another way to deliver nanoparticles to the brain without crossing the BBB is the intranasal route [363,364,365,366].

Among these, the surface functionalization of nanoparticles with specific target molecules, such as peptides or antibodies, which can recognize and bind to specific molecules overexpressed in the BBB, ensure release in the brain, and enhance drug uptake, is the most studied [364,365,366]. Indeed, glycoproteins (transferrin (tf), lactoferrin (Lf)), antibodies, apolipoproteins, peptides (cell-penetrating peptides (CPP), RVG peptides, glutathione (GSH)), vitamins (folate, thiamine), leptin, cardiolipin, mannose, are transported across the BBB by receptor-mediated transcytosis. It has been shown that modification of the surface of NPs with these ligands is the most important factor in improving BBB permeation.

Another strategy is to coat the nanoparticles with compounds such as polyethylene glycol (PEG), which confers “stealth” properties [367]. PEGylation drastically reduces opsonization, thus preventing recognition and phagocytosis by reticular system monocytes and macrophages (RES), decreasing clearance from the blood, and increasing residence time in cerebral microvessels. An overview of the main applications and benefits of nanotechnology for the treatment of AD is shown in Figure 4.

Other factors that influence the transport of nanoparticles across the BBB are their size, shape, lipophilicity, coating, and surface charge. Adjustment of these factors may produce better and more efficient nanoparticles that can penetrate the BBB and manage to escape the reticuloendothelial system for better CNS treatments. There is a wide range of functionalized nanoparticles that can be used to cross the BBB and treat central nervous system disorders. Here, however, we review recent in vitro and in vivo research on the use of nanoparticles specifically in the treatment of AD, which are reported in Table 4.

### 8.2. Nanocarriers in Alzheimer’s Disease

Nanoparticles can be designed with different types of materials, such as polymers, ceramics, carbon, metals, magnetic oxides, albumin, lipids, and liposomes. Similarly, their structure is variable, and they can be used in the form of nanofibers, nanoparticles, nanotubes, nanospheres, nanosheets, nanoflowers, nanorods, and nanogels [394,395,396].

#### 8.2.1. Metallic Nanoparticles

Metallic nanoparticles have been widely used for the treatment of various diseases, including AD. The materials used include gold, silver, selenium, iron, and cerium, and possess enhanced tunable optical properties. It is widely known that these materials have the inherent ability to cross the BBB by endocytosis involving pinocytosis and phagocytosis mechanisms [329,397]. Other interesting properties are that they can be easily conjugated with target agents and active biomolecules through H-bonds, covalent bonds, and electrostatic interactions [329,398].

In particular, gold nanoparticles (Au-NPs) are perhaps, the most important due to their low toxicity, exceptional optical properties, electrical conductance, and enhanced stability [329,399,400]. Au-NPs have been shown to inhibit the formation of Aβ aggregates, thereby reducing cognitive impairment [399]. Sanati et al. evaluated the acquisition and retention of spatial learning and memory in a model of intrahippocampal microinjection of Aβ followed by intraperitoneal (IP) administration of Au-NPs. The results showed that Au-NPs were able to improve the acquisition and retention of spatial learning and memory in rats treated with Aβ (1 μg/μL) by decreasing the time and distance to find the platform in the Morris water maze. In addition, an increase in the expression of BDNF, cAMP response element binding protein, CREB, and stromal interaction molecules (STIM1 and STIM2) was observed, favoring neuronal survival [401]. Similarly, intravenous administration of 3.3 nm L- and D-glutathione-stabilized gold nanoparticles (termed L3.3 and D3.3, respectively) was shown to be able to cross the BBB and inhibit Aβ-42 aggregation without having a toxic effect. In particular, the D3.3 enantiomer was shown to have a greater binding affinity to Aβ-42 and a higher brain biodistribution, as well as a better rescue of behavioral alterations in APPswe/PS1-dE mice [402].

Selenium has the advantage of having an exceptionally low toxicity, in addition to its antioxidant properties, it is an essential micronutrient capable of reducing oxidative stress in the brain and has neuroprotective properties. For this reason, selenium nanoparticles have received particular attention [403]. Selenium nanoparticles are known to have antioxidant properties and improve cognitive function [404]. Yin et al. synthesized a sialic acid (SA)-modified selenium (Se) nanoparticles conjugated with an alternative peptide-B6 peptide (B6-SA-SeNPs). The results showed high uptake of B6-SA-SeNPs by brain endothelial cells and that these nanoparticles could disaggregate preformed Aβ fibrils into non-toxic amorphous oligomeric forms, suggesting their potential application in the treatment of AD [405]. In another study, selenium nanoparticles containing sialic acid can cross the BBB and stop Aβ aggregation, as well as ameliorate oxidative stress [406,407]. Similarly, selenium nanoparticles modified with sialic acid and coated with peptide-B6 and epigallocate-3-gallate reduce Aβ aggregation [407].

#### 8.2.2. Polymeric Nanoparticles

Polymeric nanoparticles (pNPs) are approximately 10–100 nm in size and can be of natural or synthetic origin [408]. Examples of raw materials of natural origin for the production of pNPs are alginate, chitosan, fibrin, gelatin, collagen, and albumin. On the other hand, various polymers such as PLGA (Poly (lactic-co-glycolic acid)), PAMAM (Poly(amido)amine), PLA (poly (lactic acid)), PEG (polyethylene glycol), PCL (Polycaprolactone), and PGA (poly (glutamic acid)), are used as precursors for the production of synthetics pNPs [408]. The literature reports that pNPs can be obtained by various methods such as ionic gelation, emulsion, solvent evaporation, polymerization and spray drying, and precipitation [409]. Generally, pNPs are obtained from the polymerization of the single monomer of the aforementioned polymeric nanomaterials. pNPs are biocompatible, biodegradable, and easily eliminated from the body, which has led to their wide use in the encapsulation of a variety of drugs [410]. pNPs have been investigated in applications related to neurodegenerative diseases such as AD. Therefore, pNPs have been used to encapsulate and subsequently release bioactive agents used in the treatment of AD to improve drug retention and therapeutic efficacy. In addition, NPs have been designed to cross the BBB for targeted treatment. For this purpose, nanoparticles are conjugated with different types of ligands, such as protein-binding ligands, targeting ligands (transferrin receptor, insulin receptor, and glucose transport), ligands able to increase charge and hydrophobicity, and ligands able to increase blood circulation time [409]. Some reported examples of pNPs used in AD research are summarized in Table 5. In this table, the active agent used in the AD treatments, the type of polymeric nanoparticle used, the in vivo or in vitro model used, as well as the administration via used are reported.

#### 8.2.3. Liposomes

In the last 50 years, liposomes have been widely studied due to their remarkable advantages as drug carriers due to their structure. Liposomes are spherical vesicles with an internal aqueous cavity surrounded by a lipid bilayer, due to their nature, they are considered safe nanocarriers because they can protect the drug from enzymatic degradation, they are biocompatible and biodegradable, they show higher flexibility, they do not show immunogenicity and their toxicity is usually very low [426,427,428]. The size of these vesicles used for therapeutic use varies between 50–450 nm [428]; however, the efficacy of these drug carriers depends strictly on the physicochemical properties of their membranes, the nature of their components, their surface charge, and their lipid organization [428].

Because of their properties, liposomes have allowed the development of new methods for the targeted and specific delivery of drugs capable of crossing the BBB [397]. For example, liposomes loaded with curcumin, have been evaluated for their anti-Alzheimer effect. Studies by several research groups have shown that curcumin binds to amyloid deposits both in vitro and in vivo and not only disrupts peptide aggregation but also disaggregates preformed fibrils [429,430,431,432,433].

Mourtas et al. developed a system of nanoliposomes decorated with a curcumin derivative, designed to maintain a planar structure necessary for its interaction with Aβ. The results obtained showed an extremely high affinity for Aβ_1–42_ fibrils (1–5 nM), probably due to the occurrence of multivalent interactions. It was shown that its integrity and stability were sufficient for in vivo applications [433]. Subsequently, the same research group prepared and characterized, multifunctional liposomes that incorporated the curcumin derivative and were also decorated with a BBB transport mediator (anti-transferrin antibody). Both nanoliposomes were shown to delay the aggregation of the Aβ_1–42_ peptide, demonstrating their potential viability for application in the treatment and diagnosis of AD [431,433].

In the same sense, the development of liposomes with rivastigmine and their intranasal administration has shown that these particles can improve drug distribution and adequate retention in regions such as the hippocampus and cortex, the most affected regions by AD. At the same time, a decrease in hepatic metabolism was observed, as well as a reduction in adverse side effects, thereby increasing therapeutic efficacy [434]. Kuo et al. designed and prepared phosphatidylcholine (PC) liposomes loaded with curcumin, epigallocatechin gallate, rosmarinic acid, and quercetin with cross-linked glutathione (GSH) and apolipoprotein E (ApoE). ApoE-coated liposomes enhanced drug endocytosis in SK-N-MC cells through the low-density lipoprotein receptor (LDLR). They demonstrated that these triple target liposomes (GSH-ApoE-PC), could enhance the effect of different drugs (quercetin–curcumin, quercetin, epigallocatechin gallate, and rosmarinic acid) on damaged neural tissue in a controlled manner, favoring neuroprotection, and increasing treatment efficacy [435].

## 9. Conclusions

Despite AchEIs currently being the most widely used conventional drugs and having been in use for decades, amyloid and hyperphosphorylate tau seem to be the main components that mediate the diseases. Thus, strategies to remove or halt their formation are the most promising alternatives for treatment. Among the current and most hopeful novel proposals, vaccines, and antibody therapies seem to have the promptest success in the fight against AD. Similarly, the potential therapeutic applications of anti-inflammatory approaches and the management of dyslipidemia, along with the more recent exploration of microRNA-based therapies, have been investigated as possible avenues for the treatment of AD. While the results of such research are promising, there is, as yet, no therapeutic strategy that is demonstrably effective in preventing the development of the disease. For this reason, the targeted delivery of drugs to specific brain regions by defeating the BBB represents a significant challenge in the development of therapies for neurodegenerative diseases such as Alzheimer’s.

While nanotechnology may offer significant benefits in this area, it is essential to consider the potential for unintended consequences. The challenge of identifying safe and efficacious therapeutic options for AD persists. The combination of different strategies may prove to be a crucial step in addressing this challenge.

## Figures and Tables

**Figure 1 pharmaceutics-17-00128-f001:**
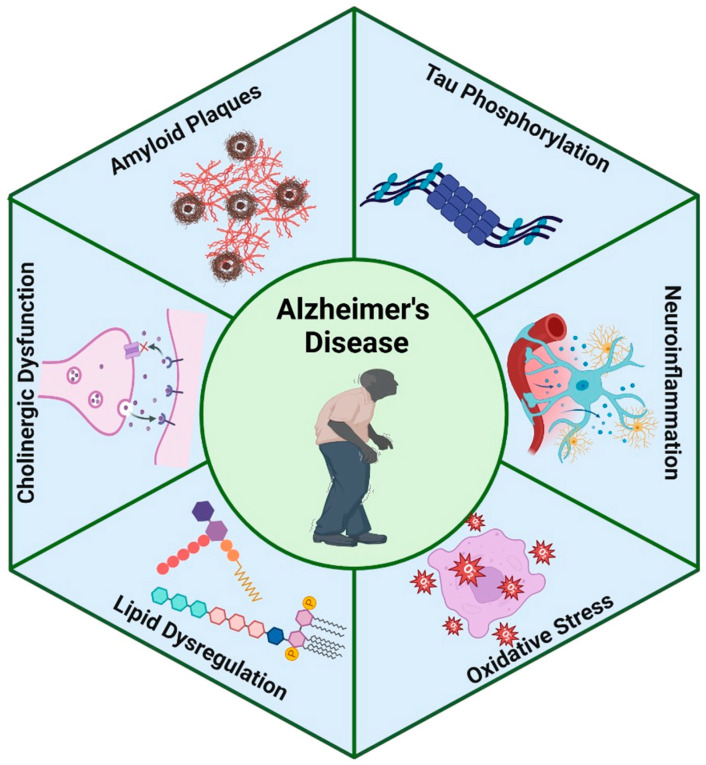
The pathophysiology of Alzheimer’s disease.

**Figure 2 pharmaceutics-17-00128-f002:**
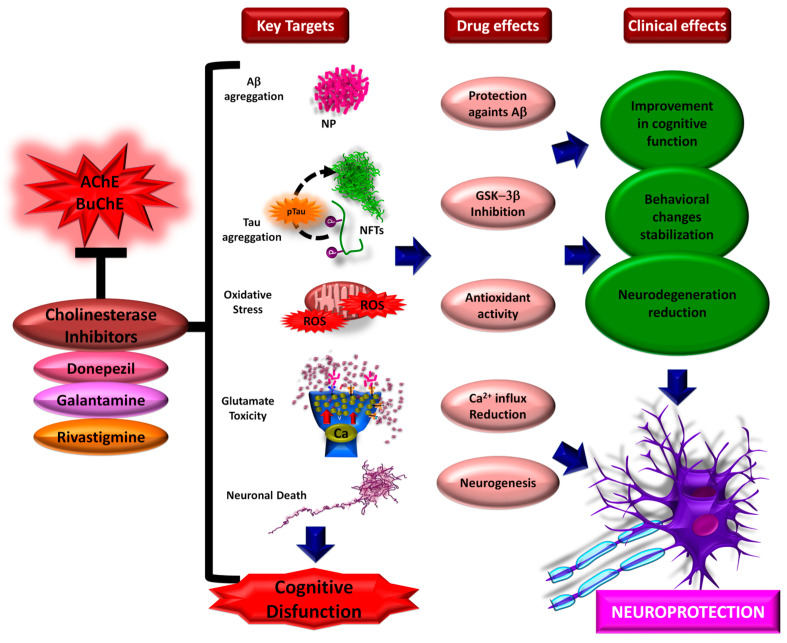
Multi-target effects of different cholinesterase inhibitors. Donepezil, galantamine, and rivastigmine show neuroprotective effects by decreasing Aβ levels, inhibiting GSK3β, and, therefore, pTau reduction. They also reduce oxidative stress and excitotoxicity levels.

**Figure 3 pharmaceutics-17-00128-f003:**
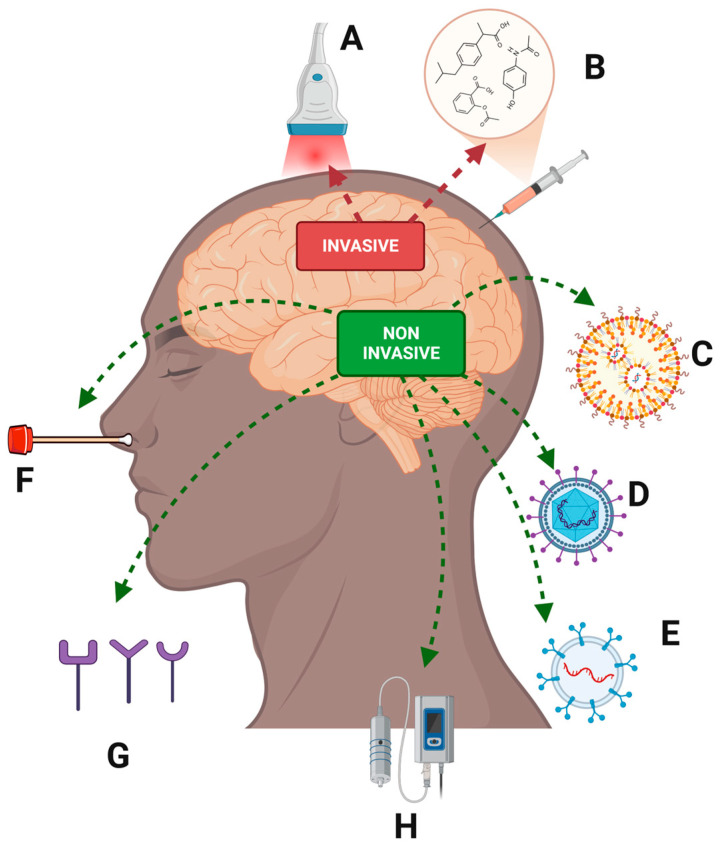
Delivery of pharmaceutical agents across the BBB into the CNS. I. INVASIVE. (**A**). The use of noxious agents, hyperosmotic solutions or ultrasound reduces endothelial cells in the brain [341]. This is achieved by disrupting tight junctions, compromising the BBB; (**B**). Injecting drugs directly into the cerebrospinal fluid is a way of delivering pharmaceutical agents to the central nervous system. This can be done through a lumbar puncture or an implanted device [342,343]. II. NON-INVASIVE: (**C**). A drug may be combined with lipoids for brain delivery. This approach may be constrained by the trade-off between increased lipophilicity and increased biodistribution. This technique necessitates that the treatment emulates endogenous ligands. Glycosylation enhances drug and peptide transport to the brain while improving stability. Other processes, such as cyclization, halogenation, methylation, or unnatural linkers, can be used to modify BBB-crossing drugs [344,345]; (**D**). Viral vectors can cross the BBB by transcytosis or temporary BBB disruption. Transcytosis occurs via receptor-mediated vascular endothelial cells of the brain. The temporary disruption of the BBB permits the vector to gain access to the interstitial regions of the CNS via paracellular transport [346,347]; (**E**). Surface modifications could help exosomes cross the BBB. Exosomes can be functionalized with biomolecules and polymers without affecting their activity. Chemical modifications are attractive due to synthesis, yield, and chemical reaction [348,349,350]; (**F**). Intranasal (IN) administration delivers drugs directly to the CNS via the nasal cavity. This method delivers the agent directly to the CNS, reducing exposure and adverse effects. Therapeutic agents reach the central nervous system rapidly [351,352,353]; (**G**). Four different adenosine receptors (A1, A3, A2A, and A2B) can regulate the BBB permeability. A2A regulates BBB permeability via actin-cytoskeletal reorganization, affecting tight junctions [354,355,356,357]; (**H**). Microbubble-enhanced diagnostic ultrasound (MEUS) facilitates drug passage through the BBB in glioma patients. Principal proteins at the BBB transluminal junctions are claudins, occludin, and JAMs. The use of ultrasound and microbubbles has been shown to suppress the expression of these proteins at the translaminar junctions, thereby opening the BBB in a relatively short period of time without causing damage to normal brain tissue [358].

**Figure 4 pharmaceutics-17-00128-f004:**
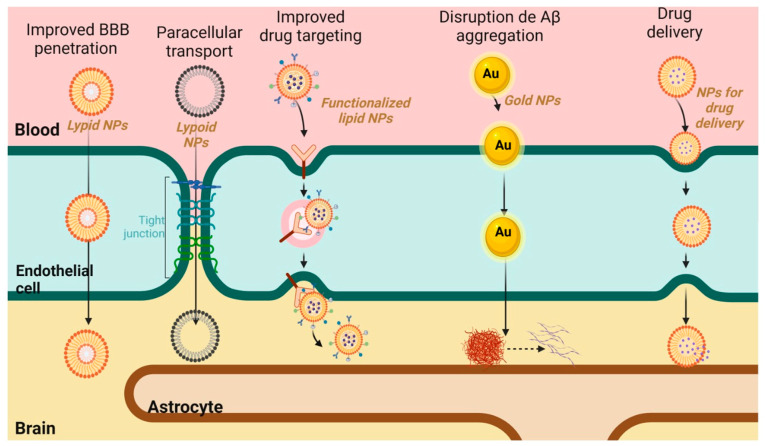
Applications of nanotechnology in the treatment of AD: Nanoparticles, due to their small size, can more effectively cross the BBB. Lipid-soluble nanoparticles, such as liposomes, can be used to transport drugs through the BBB, thus increasing the drugs’ bioavailability. Metallic nanoparticles, such as gold nanoparticles, have been used to inhibit the formation of Aβ aggregates, inhibiting the formation of neuritic plaques. Covalent bonding of drugs with specific nanoparticles can improve the active uptake of the bonded drugs by the endothelial cells, increasing their uptake into the brain and facilitating drug targeting. Created with BioRender.com. BBB, blood–brain barrier; APP, amyloid precursor protein; Aβ, Amyloid beta; Au, Gold; C, Carbon.

**Table 1 pharmaceutics-17-00128-t001:** Transgenic models designed from mutations in *APP* (a), and models designed from mutations in *APP* and *PSEN1*/2 (b).

Animal Models	Mutation	Phenotype/Pathology	Reference
(a) *APP* Mutations Animal Models			
PDAPP	Indiana mutation (V717F)	Amyloid deposits (3–9 months), neuroinflammation, behavioral impairments (3 months)	[34,35]
Tg2576	APP695 gene with the Swedish mutation (KM670/671NL)	Amyloid deposits (11–13 months), neuroinflammation, behavioral impairments (9 months), neuronal loss	[36,37]
APP23	APP751 gene with the Swedish mutation (KM670/671NL)	Amyloid deposits (6 months), neuroinflammation, behavioural impairments (3 months), neuronal loss	[38,39,40,41]
J20	*APP* gene with Swedish mutation (KM670/671NL) and the Indiana mutation (v717F)	Amyloid deposits (7–9 months), neuroinflammation, behavioural impairments (4 months), neuronal loss	[42,43,44]
TgCRND8	APP695 gene with Swedish (KM670/671NL) and the Indiana (V717F) mutation using the hamster PrP gene promoter	Amyloid deposits (3 months), neuroinflammation, behavioural impairments (3 months)	[45,46,47,48]
APPN^L_G_F^ Knock-In	Swedish mutation (KM670/671NL), the arctic mutation (E693G)	Amyloid deposits (2 months), neuroinflammation, behavioural impairments (6–9 months)	[49,50,51,52,53]
**(b) *APP* + *PSEN* Mutations Animal Models**			
APP/PS1 (Tg2576xPS1)	*APP* gene with Swedish mutation (KM670/671NL) and M146L mutation in the *PSEN1* gene	Amyloid deposits (6 months), neuroinflammation, behavioural impairments (3–6 months)	[54,55,56,57]
APP/PS1 (APPswe/PSEN1ΔE9)	*APP* gene with Swedish mutation (KM670/671NL) and deletion of exon 9 in *PSEN1*	Amyloid deposits (4 months), neuroinflammation, behavioural impairments (8 months), neuronal loss	[58,59,60,61,62,63,64,65,66]
APP_SL_/PS1	*APP* gene Swedish mutation (KM670/671NL) and the London mutation (V717I), and *PSEN1* gene with the M146L mutation	Amyloid deposits (2.5 months), neuroinflammation, behavioural impairments (9 months), neuronal loss	[67,68,69,70,71,72]
PS2APP	*APP* gene with the Swedish mutation (KM670/671NL) and *PSEN2* gene with the N141I mutation	Amyloid deposits (5–6 months), neuroinflammation, behavioural impairments (7–8 months)	[73,74,75,76]
APP_SL_PS1 Knock-In	*APP* gene Swedish mutation (KM670/671NL), London mutation (V717I), and carry within the mouse endogenous *PSEN1* gene, the M233T/L235P mutations	Amyloid deposits (2–3 months), neuroinflammation, behavioural impairments (6 months), neuronal loss	[77,78,79,80,81]
5xFAD	*APP* gene with the Swedish (KM670/671NL), London (V717I) and Florida (I716V) mutations and the *PSEN1* gene with the M146L and the L286V mutations	Amyloid deposits (2 months), neuroinflammation, behavioural impairments (1–4 months), neuronal loss	[82,83,84,85,86,87,88,89,90]

**Table 2 pharmaceutics-17-00128-t002:** Immunotherapy targeting Aβ.

Immunotherapy	Drug	Population	Phase	Reference
Active immunotherapy	AN1792	Mild to moderate AD	II	[190]
	CAD106	AD patients, people at high risk of developing late-onset AD	II/III	[191,192,193]
	UB-311	Mild to moderate AD	II	[194,195]
	ABvac40	Mild to moderate AD	II	[196]
Passive immunotherapy	Solanezumab	Mild to moderate AD, prodromal AD, participants at risk of memory loss	III	[197,198]
	Gantenerumab	Mild AD, prodromal to mild AD, early AD	III	[123,199,200]
	Aducanumab	Early AD	III	[201,202,203]
	Crenezumab	Prodromal to mild AD	III	[204]
	Lecanemab	Early AD, preclinical AD	III	[205]
	Donanemab	Early symptomatic AD, preclinical AD	III	[206]

**Table 3 pharmaceutics-17-00128-t003:** Immunotherapy targeting tau.

Immunotherapy	Drug		Population	Phase	Reference
Active immunotherapy	AADvac1	Tau-directed vaccine	Mild to moderate AD, prodromal AD, Participants at risk of memory loss	I/II	[221,222]
	ACI-35	Liposomal vaccine	Early AD	I	[223]
Passive Immunotherapy	E2814	Mid-domain tau antibody	Mild AD, moderate AD	II	[224]
	JNJ-63733657	Mid-domain tau antibody	Prodromal AD, mild AD	I/II	[225,226]
	Bepranemab	Mid-domain tau antibody	Prodromal AD, mild AD	II	[227,228]
	PRX005	Mid-domain tau antibody	Healthy participants	I	[229]
	Lu AF87908	C-terminal ptau antibody	Early AD	I	[230,231,232]
	PNT00I	C-terminal ptau antibody	AD	I	[233,234]

**Table 4 pharmaceutics-17-00128-t004:** The in vitro and in vivo reports on the use of functionalized nanoparticles in AD. Abbreviations: NPs (nanoparticles), PLG (polyethylene glycol), CD-MOF (cyclodextrin-based metal–organic frameworks), ApoE (apolipoprotein E), Au (gold).

In Vitro					
Nanoparticle	Ligand	Therapeutic Molecule	Mechanism to Cross the BBB	Model	Reference
Lipid NPs	Peptide 22	Tegaserod	Mediated by receptor	Hemolysis assay	[368]
Liposomes	Transferrin	Vitamin B12	Mediated by receptor	Lyophilized Aβ_1–42_ monomers	[369]
Lipid NPs	RVG29 peptide	Quercetin	Mediated by receptor	hCMEC/D3 cells	[370]
Polymeric NPs	Tween 80	Rhynchophylline	Mediated by receptor	Mouse brain endothelial cells (bEnd.3)	[371]
Lipid NPs	Transferrin	Quercetin	Mediated by receptor	hCMEC/ D3 cell line	[372]
Liposomes	Transferrin	Caffeic acid	Mediated by receptor	Human Aβ_1−42_	[373]
Liposomes	Transferrin	Gallic acid	Mediated by receptor	Human Aβ_1−42_	[374]
Lipid NPs	ApoE	Donepezil	Mediated by receptor	RBEC, hCMEC/D3, and SH-SY5Y cells	[375]
In Vivo					
Layered double hydroxide NPs	Ang2 and RVG29	Rutin		APP/PS1 and Tau.P301S AD mouse model	[376]
PLG NPs	Phenylalanine dipeptide	Morin hydrated		Wistar rats	[377]
Liposomes	Glucose-mannose	Curcumin	Mediated by receptor	APP/PS-1 mice	[378]
CD-MOF	Lactoferrin	Huperzine A	Intranasal	PC12 cells and rats	[379]
Lipid NPs	Rabbit virus glycoprotein	miR-137-3p	Virus-mediated	Neuroblastoma cells and AD mouse model	[380]
Calcium-doped mesoporous silica NPs	Polysorbate-80	Rivastigmine	Mediated by receptor	Rats	[381]
PLG NPs	Polysorbate-80	Thymoquinone	Mediated by receptor	Streptozotocin-(STZ)-induced Alzheimer’s mice	[382]
Albumin NPs	T807 and triphenylphosphine	Curcumin	Mediated by receptor	Endothelial cells (BMECs) /AD model mice	[383]
Liposomes	Mannose, and cell-penetrating peptides	Brain-derived neurotrophic factor (BDNF)	Mediated by receptor	Transgenic APP/PS1 AD mice	[384]
Liposomes	Glucose transporter-cell-penetrating peptides	VGF	Mediated by receptor	in vitro BBB/mice	[385]
Polymeric NPs	RVG29	shRNA and epigallocatechin-3-gallate	Mediated by receptor	APPswe/PS1dE9 doubletransgenic mice	[386]
Chitosan-coated solid lipid NPs	Compritol/polysorbate 80	Ferulic acid	Intranasal	Goat nasal mucosa/AD-induced rats	[387]
Liposomes	Intranasal	α-tocopherol and donepezil hydrochloride	Intranasal	Rats	[388]
Liposomes	Intranasal	Hydroxy-α-sanshool	Intranasal	Rat nasal mucosa/AD mice	[389]
Liposomes	c(RGDyK) cyclic peptide	Ammonium antidotes (HI-6)	Intranasal	BMECs monolayer culture in vitro BBB model/rats	[390]
Liposomes	Transferrin	Osthole	Mediated by receptor	APP-SH-SY5Y cells and APP/PS-1 mice	[391]
Liposomes	Transferrin	Pep63		APP/PS1 mice	[392]
Liposomes	ApoE3	Rivastigmine	Mediated by receptor	AChE assay by Ellman’s method	[393]
Lipid NPs	Rabies virus glycoprotein	miR-137-39	intranasal	Neuroblastoma cells and a mouse model of AD	[380]
Mesoporous silica NPs	Polysorbate-80	Rivastigimine		rats	[381]
Gold NPs	Polyethylene glycol	Au-G		P301L mice	[367]

**Table 5 pharmaceutics-17-00128-t005:** Polymeric nanoparticles for targeted drug delivery in the treatment of Alzheimer’s disease, in vivo or in vitro study and animal model used.

Active Agent	Type of Polymeric Nanoparticle	Used Model In Vitro or In Vivo	Administration via	Reference
Anti-TRIAL monoclonal antibody	PLGA	(1) Murine macrophage RAW 264.7 cell line (2) 3xTg-AD mice	Nasal	[411]
Curcumin	Aβ-PEG-LysB	APP/PS1 model mice	Nasal	[261]
Romidepsin and Metformin	Poloxamer	Streptozocin-mediated AD model	Intravenous	[262]
Donepezil	Chitosan	Wistar rats	Intranasal	[412]
Rivastigmine and Quercetin	PCL-PEG-PCL	Scopolamine-induced Wistar rats	Intraperitoneal	[413]
Pioglitazone	PLGA-PEG	Ex vivo permeation studies using buccal, sublingual, nasal and intestinal mucosa		[414]
Epigallocatechin-3-gallate and β-site amyloid precursor protein cleaving enzyme 1 antisense shRNA-encoded plasmid	PEGilated PLGA	APP/PS1 mice	Intravenous	[386]
4-phenyltellanyl-7-chloroquinoline	PCL	(1) AD model in transgenic *Caenorhabditis elegans* expressing human Aβ_1–42_ in their body-wall muscles (2) Swiss mice injected with Aβ_25–35_	Intragastrically administered via oral gavage	[415]
Nattokinase	PLGA	In vitro anti-amyloid activity		[416]
Withaferin-A	PLGA	In vitro drug release		[417]
Curcumin	PLGA	Hippocampal cell cultures	In vitro Aβ pathology	[418]
*Frankincense*	PMBC	Scopolamine-treated Wistar rats	Intraperitoneally	[419]
Meloxicam	PCL	aβ(25–35) peptide-induced damage in mice	Intracerebroventricular	[420]
Rosmarinic acid and curcumin	PAAM-CL-PLGA	SK-N-MC cells	In vitro study of expression of phosphorylated mitogen-activated protein kinase, phosphorylated p38 and phosphorylated tau protein	[421]
Curcumin	PCL	Zebrafish	Scopolamine-induced zebrafish	[422]
Zn^2+^	PLGA	Wild-type and APP23 mice	Intraperitoneal	[423]
Memantine	PEG-PLGA	Scopolamine-induced mice	Intrathecal route	[424]
High-density lipoproteins	Chitosan	APP/PS1 transgenic AD mice	Intranasal	[425]

## Data Availability

Not applicable.

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
