# Peer review of "Evolution of Alzheimer’s Disease Therapeutics: From Conventional Drugs to Medicinal Plants, Immunotherapy, Microbiotherapy and Nanotherapy"

_pharmaceutics, 2025, doi:10.3390/pharmaceutics17010128_

Round 1
Reviewer 1 Report
Comments and Suggestions for Authors
Major comments:
1) Abstract and other sections should be aligned according to the template (Justify, not aligned to the left). Please refer to the Instructions to authors (https://www.mdpi.com/journal/pharmaceutics/instructions) and MS Word template.
2) Although Tables and Figures are provided by authors as separate files, they also should be integrated into the manuscript.
3) Throughout the manuscript I noticed that there is an interchange between Tau and tau, please be consistent in naming. I would suggest to adhere to Tau or p-Tau (with a capital T).
4) Please add a short section on Proteolysis targeting chimera (PROTAC) technology which is now regarded as a promising approach for selectively degrading dysfunctional proteins such as phosphorylated Tau in AD.
Minor comments:
Overall remarks:
1) There should be no period (.) in the title as well as in titles of sections and subsections.
2) Formatting issues with Abstract and Keywords are intended as well as different type fonts are used in the manuscript.
3) Names of drugs that are on the market or potent compounds that are in trials should be named with the first capital letter. For example, Donepizil, Memantine, etc.
4) Is there a particular reason: Conflicts of Interest: “The authors declare no conflicts of interest.” is in quotation marks?
5) Line 348: here, and throughout the text there is seem to be a salad of naming beta-amyloid. I would suggest introduce this term in the Introduction section as Aβ and adhere to it.
Line by line remarks:
Line 32: “This review provides a comprehensive overview…” – awkward wording, needs a rewrite.
Line 75: A run-on sentence: “AD's etiology is complex; several theories attempt to explain its causes”, please consider a rewrite.
Line 127: what is AβPP?
Line 242: Food and drug adminitration (FDA): a typo in naming and it should Food and Drug Administration.
Line 287 and 006 (pg. 21): is there a particular reason that Aβ is in bold?
Line 320: missing period at the end of the sentence and closing bracket is in bold.
Line 761, 767, 989 and more: missing (’) in Alzheimer’s disease.
Line 768: typo in “oxidacion”.
Line 108 (pg. 23): “Here, the development of antibodies and vaccines are hopefulness choices.” – awkward wording, needs a rewrite.
Comments on the Quality of English LanguageMajor comments:
1) Abstract and other sections should be aligned according to the template (Justify, not aligned to the left). Please refer to the Instructions to authors (https://www.mdpi.com/journal/pharmaceutics/instructions) and MS Word template.
2) Although Tables and Figures are provided by authors as separate files, they also should be integrated into the manuscript.
3) Throughout the manuscript I noticed that there is an interchange between Tau and tau, please be consistent in naming. I would suggest to adhere to Tau or p-Tau (with a capital T).
4) Please add a short section on Proteolysis targeting chimera (PROTAC) technology which is now regarded as a promising approach for selectively degrading dysfunctional proteins such as phosphorylated Tau in AD.
Minor comments:
Overall remarks:
1) There should be no period (.) in the title as well as in titles of sections and subsections.
2) Formatting issues with Abstract and Keywords are intended as well as different type fonts are used in the manuscript.
3) Names of drugs that are on the market or potent compounds that are in trials should be named with the first capital letter. For example, Donepizil, Memantine, etc.
4) Is there a particular reason: Conflicts of Interest: “The authors declare no conflicts of interest.” is in quotation marks?
5) Line 348: here, and throughout the text there is seem to be a salad of naming beta-amyloid. I would suggest introduce this term in the Introduction section as Aβ and adhere to it.
Line by line remarks:
Line 32: “This review provides a comprehensive overview…” – awkward wording, needs a rewrite.
Line 75: A run-on sentence: “AD's etiology is complex; several theories attempt to explain its causes”, please consider a rewrite.
Line 127: what is AβPP?
Line 242: Food and drug adminitration (FDA): a typo in naming and it should Food and Drug Administration.
Line 287 and 006 (pg. 21): is there a particular reason that Aβ is in bold?
Line 320: missing period at the end of the sentence and closing bracket is in bold.
Line 761, 767, 989 and more: missing (’) in Alzheimer’s disease.
Line 768: typo in “oxidacion”.
Line 108 (pg. 23): “Here, the development of antibodies and vaccines are hopefulness choices.” – awkward wording, needs a rewrite.
Author Response
Response to Reviewer 1
- Summary
Thank you very much for taking the time to review this manuscript. Below you will find detailed responses and corresponding corrections highlighted in yellow throughout the manuscript.
- Major comments:
Comments 1: Abstract and other sections should be aligned according to the template (Justify, not aligned to the left). Please refer to the Instructions to authors (https://www.mdpi.com/journal/pharmaceutics/instructions) and MS Word template
Response 1: Thanks for pointing this out and the text has been aligned according to the template.
Comments 2: Although Tables and Figures are provided by authors as separate files, they also should be integrated into the manuscript.
Response 2: Thank you for the comment. We have added the figures and tables to the manuscript as suggested.
Comments 3: Throughout the manuscript I noticed that there is an interchange between Tau and tau, please be consistent in naming. I would suggest to adhere to Tau or p-Tau (with a capital T).
Response 3: We agree with this comment. We have therefore made the suggested changes.
Comments 4: Please add a short section on Proteolysis targeting chimera (PROTAC) technology which is now regarded as a promising approach for selectively degrading dysfunctional proteins such as phosphorylated Tau in AD
Response 4: Thanks for the suggestion. We have added a short section on Proteolysis Targeting Chimera (PROTAC) technology. Hopefully this will live up to expectations.
- Minor comments:
Overall remarks:
Comments 1: There should be no period (.) in the title as well as in titles of sections and subsections
Response 1: We agree with this comment. The points have been removed as suggested.
Comments 2: Formatting issues with Abstract and Keywords are intended as well as different type fonts are used in the manuscript.
Response 2: Thanks for pointing this out; we have corrected what the reviewer pointed out.
Comments 3: Names of drugs that are on the market or potent compounds that are in trials should be named with the first capital letter. For example, Donepizil, Memantine, etc.
Response 3: We appreciate your comment. However, we believe that the names of donepezil, memantine, etc. should be kept in lower case since this is the active compound.
Comments 4: Is there a particular reason: Conflicts of Interest: “The authors declare no conflicts of interest.” is in quotation marks?
Response 4: There is no particular reason, however the quotation marks were removed.
Comments 5: Line 348: here, and throughout the text there is seem to be a salad of naming beta-amyloid. I would suggest introduce this term in the Introduction section as Aβ and adhere to it.
Response 5: Thank you for your comment. We have introduced the term Aβ as suggested and made corrections throughout the text.
- Line by line remarks:
Line 32: “This review provides a comprehensive overview…” – awkward wording, needs a rewrite.
Thank you for your comment. We have changed the wording.
Line 75: A run-on sentence: “AD's etiology is complex; several theories attempt to explain its causes”, please consider a rewrite.
We have made changes to the wording.
Line 127: what is AβPP?
The error has been corrected.
Line 242: Food and drug adminitration (FDA): a typo in naming and it should Food and Drug Administration.
We have modified the error.
Line 287 and 006 (pg. 21): is there a particular reason that Aβ is in bold?
No, there is nothing in particular, it was a mistake, which was corrected.
Line 320: missing period at the end of the sentence and closing bracket is in bold.
The error has been corrected.
Line 761, 767, 989 and more: missing (’) in Alzheimer’s disease.
We have corrected the error.
Line 768: typo in “oxidacion”.
We have corrected the error.
Line 108 (pg. 23): “Here, the development of antibodies and vaccines are hopefulness choices.” – awkward wording, needs a rewrite.
The wording has been changed.
Reviewer 2 Report
Comments and Suggestions for Authors
In this manuscript, the authors provide a comprehensive elucidation of the pathogenesis of Alzheimer's disease (AD) and present a detailed overview of therapeutic strategies explored for its treatment. The manuscript highlights precise mechanisms of drug action utilized in AD patients, points to using monoclonal antibodies to eliminate toxic proteins implicated in AD, as well as the therapeutic potential of medicinal plants and recent advancements in nanotechnology, which show promise in improving the care and management of AD.
I have one question for the authors:
Given that lipid metabolism is implicated in AD pathology, in your opinion, could lipidomic analysis of serum via LC/MS identify specific lipids as potential biomarkers for this disease? Moreover, might these lipid biomarkers serve as therapeutic targets for future interventions?
Author Response
Response to Reviewer 2
- Summary
We would like to thank of the reviewer for the time and effort they have put into the review of our manuscript. Below is the answer to your question, in the hope that it is in line with your expectations.
Comments 1: In this manuscript, the authors provide a comprehensive elucidation of the pathogenesis of Alzheimer's disease (AD) and present a detailed overview of therapeutic strategies explored for its treatment. The manuscript highlights precise mechanisms of drug action utilized in AD patients, points to using monoclonal antibodies to eliminate toxic proteins implicated in AD, as well as the therapeutic potential of medicinal plants and recent advancements innanotechnology, which show promise in improving the care and management of AD.
I have one question for the authors:
Given that lipid metabolism is implicated in AD pathology, in your opinion, could lipidomic analysis of serum via LC/MS identify specific lipids as potential biomarkers for this disease? Moreover, might these lipid biomarkers serve as therapeutic targets for future interventions?
Response 1: Lipids are fundamental components of cellular structure and function, particularly in the brain, which is one of the most lipid-rich organs. Therefore, alterations in lipid homeostasis can lead to irreversible changes that contribute to the development of several neurodegenerative diseases, such as Alzheimer's disease. We also know that abnormal accumulation of lipids in blood vessels is associated with recurrent and silent strokes, which ultimately lead to an increased risk of dementia.
Lipidomics is a powerful tool that will undoubtedly allow us to study the different types of lipids in the periphery and to use them as markers in the preclinical stages of Alzheimer's disease. The specific analysis of this type of marker will not only allow us to identify lipid profiles associated with different stages of the disease, but could also be used to develop new therapies that modify this lipid profile before neuropathological changes occur. Studies are currently underway that demonstrate the importance and critical role of plasma lipidomic analysis in facilitating early diagnosis and the development of new therapeutic strategies [1, 2].
- Otoki, Y.; Yu, D.; Shen, Q.; Sahlas, D. J.; Ramirez, J.; Gao, F.; Masellis, M.; Swartz, R. H.; Chan, P. C.; Pettersen, J. A.; Kato, S.; Nakagawa, K.; Black, S. E.; Swardfager, W.; Taha, A. Y., Quantitative Lipidomic Analysis of Serum Phospholipids Reveals Dissociable Markers of Alzheimer's Disease and Subcortical Cerebrovascular Disease. J Alzheimers Dis 2023, 93, (2), 665-682.
- Krokidis, M. G.; Pucha, K. A.; Mustapic, M.; Exarchos, T. P.; Vlamos, P.; Kapogiannis, D., Lipidomic Analysis of Plasma Extracellular Vesicles Derived from Alzheimer's Disease Patients. Cells 2024, 13, (8).